# Conifer-killing bark beetles locate fungal symbionts by detecting volatile fungal metabolites of host tree resin monoterpenes

**Dineshkumar Kandasamy**[1,2¤]*, **Rashaduz Zaman**[1], **Yoko Nakamura**[3,4], **Tao Zhao**[5], **Henrik Hartmann**[6], **Martin N. Andersson**[2,7], **Almuth Hammerbacher**[8], **Jonathan Gershenzon** [1]*

**1** Department of Biochemistry, Max Planck Institute for Chemical Ecology, Jena, Germany, **2** Max Planck Center for next Generation Insect Chemical Ecology (nGICE), Department of Biology, Lund University, Lund, Sweden, **3** Department of Natural Product Biosynthesis, Max Planck Institute for Chemical Ecology, Jena, Germany, **4** Research Group Biosynthesis/NMR, Max Planck Institute for Chemical Ecology, Jena, Germany, **5** School of Science and Technology, Örebro University, Örebro, Sweden, **6** Department of Biogeochemical Processes, Max Planck Institute for Biogeochemistry, Jena, Germany, **7** Department of Biology, Lund University, Lund, Sweden, **8** Department of Zoology and Entomology, Forestry and Agricultural Biotechnology Institute, University of Pretoria, Pretoria, South Africa

¤ Current address: Department of Biology, Lund University, Lund, Sweden
* dineshkumar.kandasamy@biol.lu.se (DK); gershenzon@ice.mpg.de (JG)

**Data Availability Statement:** All relevant data are either within the manuscript and its supporting information files, or available online https://doi.org/10.6084/m9.figshare.21692156.v1.

## Abstract

Outbreaks of the Eurasian spruce bark beetle (*Ips typographus*) have decimated millions of hectares of conifer forests in Europe in recent years. The ability of these 4.0 to 5.5 mm long insects to kill mature trees over a short period has been sometimes ascribed to two main factors: (1) mass attacks on the host tree to overcome tree defenses and (2) the presence of fungal symbionts that support successful beetle development in the tree. While the role of pheromones in coordinating mass attacks has been well studied, the role of chemical communication in maintaining the fungal symbiosis is poorly understood. Previous evidence indicates that *I. typographus* can distinguish fungal symbionts of the genera *Grosmannia*, *Endoconidiophora*, and *Ophiostoma* by their de novo synthesized volatile compounds. Here, we hypothesize that the fungal symbionts of this bark beetle species metabolize spruce resin monoterpenes of the beetle's host tree, Norway spruce (*Picea abies*), and that the volatile products are used as cues by beetles for locating breeding sites with beneficial symbionts. We show that *Grosmannia penicillata* and other fungal symbionts alter the profile of spruce bark volatiles by converting the major monoterpenes into an attractive blend of oxygenated derivatives. Bornyl acetate was metabolized to camphor, and α- and β-pinene to *trans*-4-thujanol and other oxygenated products. Electrophysiological measurements showed that *I. typographus* possesses dedicated olfactory sensory neurons for oxygenated metabolites. Both camphor and *trans*-4-thujanol attracted beetles at specific doses in walking olfactometer experiments, and the presence of symbiotic fungi enhanced attraction of females to pheromones. Another co-occurring nonbeneficial fungus (*Trichoderma* sp.) also produced oxygenated monoterpenes, but these were not attractive to *I. typographus*. Finally, we show that colonization of fungal symbionts on spruce bark diet stimulated beetles to make tunnels into the diet. Collectively, our study suggests that the blends of oxygenated

**Funding:** Financial support was provided by the Max Planck society to DK, RZ, YN, HH, AH and JG. MNA acknowledges funding from the Swedish Research Council FORMAS (grants #217-2014-689 and #2018-01444) the Crafoord foundation, the Royal Physiographic Society in Lund, and the Foundation in Memory of Oscar and Lily Lamm. The funders had no role in study design, data collection and analysis, decision to publish and writing.

**Competing interests:** The authors have declared that no competing interests exist.

**Abbreviations:** DMSO, dimethyl sulfoxide; FID, flame ionization detector; GC-FID, gas chromatography with flame ionization detection; GC–MS, gas chromatography–mass spectrometry; MT, monoterpene; OSN, olfactory sensory neuron; PDA, potato dextrose agar; PDMS, polydimethylsiloxane; SBA, spruce bark agar; SSR, single sensillum recording.

metabolites of conifer monoterpenes produced by fungal symbionts are used by walking bark beetles as attractive or repellent cues to locate breeding or feeding sites containing beneficial microbial symbionts. The oxygenated metabolites may aid beetles in assessing the presence of the fungus, the defense status of the host tree and the density of conspecifics at potential feeding and breeding sites.

## Introduction

Many interactions between insects and their host plants are known to be mediated by volatile organic compounds. In contrast, volatile signals between herbivorous insects and their symbiotic microbes have been less studied, aside from a few well-known examples including ambrosia beetles, termites, and the vinegar fly *Drosophila melanogaster* [1–4]. Yet, such signals could be as critical for insect fitness as their response to host plant cues. In some insect–microbe symbioses, microbes transform host plant metabolites creating volatile signals that are used by insects for food or brood site selection [3–6]. For example, yeasts vectored by *D. melanogaster* metabolize dietary phenolic antioxidants and release volatile phenolics that attract both larvae and adults to feed on antioxidant-rich foods [7]. Nevertheless, there is still little information about how microbial transformation of host plant chemicals influences insect–microbe symbioses, and whether the resulting metabolites represent honest signals of partner benefits.

Microbial symbioses are especially characteristic of wood-boring insects such as bark and ambrosia beetles. Bark beetles have captured much attention recently because of their large-scale outbreaks in many parts of the world. In Europe, for example, the Eurasian spruce bark beetle (*Ips typographus*) has killed millions of hectares of spruce stands as a result of major population bursts, likely in response to global warming and management practices that increase forest vulnerability to epidemic outbreaks [8–12]. This bark beetle feeds and reproduces in the phloem tissue of trees, which contains high levels of terpene and phenolic defense chemicals [8,13]. This insect overcomes its unfavorable environment by mass attacks and by introducing a suite of microbes into the host tree, including ectosymbiotic ophiostomatoid fungi, primarily *Grosmannia penicillata*, *Leptographium europhioides*, *Endoconidiophora polonica*, and *Ophiostoma bicolor* that cause blue staining of infected wood [14–18]. The ecological relationships between these fungal symbionts and *I. typographus* have not been extensively studied. Although they grow outside the insect, as necrotrophic fungi, they may serve as mutualists by exhausting host tree defenses, metabolizing host defense compounds, and providing nutritional benefits to larvae and adults [19–21]. However, some ophiostomatoid fungal associates of bark beetles may also act as commensals or antagonists to beetle brood development [22].

Conifer oleoresins are a formidable defense against insects and pathogens, as they can poison and physically entrap invaders [23–26]. However, the volatile fraction of the resin, especially the monoterpenes, also plays a central role in the colonization of host trees by bark beetles [24,27,28]. After locating a suitable tree, pioneer male *I. typographus* oxidize the dominant host monoterpene α-pinene to *cis*-verbenol, which is used as an aggregation pheromone in combination with the de novo produced 2-methyl-3-buten-2-ol to attract conspecifics for a mass attack [29–31]. In addition to bark beetle pheromones, several other oxygenated monoterpenes such as terpinen-4-ol, camphor, *trans*-4-thujanol, and borneol have also been detected at the entrance holes of *I. typographus* galleries [32–35]. Additionally, the phloem colonized by ophiostomatoid fungi around these galleries also produces large amounts of oxygenated monoterpenes compared to galleries without evident fungal growth [33]. Only a few studies have investigated the changes in the oleoresin composition of conifers within the

necrotic lesions caused by fungi [36,37]. While these studies showed that fungal infection changed the composition of oleoresins, the presence of oxygenated monoterpenes and their ecological roles were not investigated. In our previous work, we showed that *I. typographus* utilizes de novo synthesized fungal volatiles to maintain its association with specific beneficial symbionts and also to avoid saprophytes [38]. However, it is unknown which volatiles are produced by these fungi when they colonize their native substrate i.e., the phloem and sapwood of the tree.

In this study, we investigated the volatile compounds emitted when fungal symbionts of *I. typographus* infect the excised bark of their Norway spruce (*Picea abies*) host trees. We show that these fungi alter the volatile monoterpene composition of spruce bark and demonstrate, using single sensillum recordings (SSRs), that adult *I. typographus* can perceive the fungal-produced monoterpenes and are attracted to these compounds in behavioral bioassays. Our results imply that bark beetles respond to fungal symbiont biotransformation products of host tree metabolites and employ them to identify suitable sites for feeding and breeding.

## Results

### *Ips typographus* is attracted to volatiles from fungal symbionts on a spruce bark medium

Adult *I. typographus* beetles were strongly attracted to volatiles emitted by their symbiotic fungus *G. penicillata* grown on both potato dextrose agar (PDA) or spruce bark agar (SBA) compared to the respective fungus-free agar controls in laboratory trap bioassays (Fig 1) (PDA, $z = 3.34$, $p = 0.001$; SBA, $z = 2.83$, $p = 0.005$, Wilcoxon's test). Further, bark beetles showed a stronger attraction towards *G. penicillata* grown on SBA over the same fungus grown on PDA (Fig 1D) ($z = 4.28$, $p < 0.001$, Wilcoxon's test). Volatiles from several other bark beetle fungal symbionts, such as *E. polonica* and *L. europhioides*, grown on SBA were also attractive to adult beetles, although not all bark beetle-associated fungi, nor an antagonistic, nonsymbiotic saprophyte (*Trichoderma* sp.), tested in this way emitted attractive volatile blends (S1 Fig) [39].

### Symbiotic fungi alter the volatile profile of the bark

The headspace volatiles of *P. abies* bark inoculated with *I. typographus* fungal symbionts were distinct from those of fungus-free bark 4 d after inoculation when analyzed using gas chromatography with flame ionization detection (GC-FID) and gas chromatography–mass spectrometry (GC–MS) (S2–S7 Tables). For *G. penicillata*-inoculated bark, PCA analysis revealed that nearly 68% of the variation in the volatile profiles was explained by the first two principal components (Fig 2A).

Over a time course of 4, 8, 12, and 18 d post-inoculation with *G. penicillata* on detached bark plugs (Fig 2B and S3 and S2, S3, and S5 Tables), 79 compounds comprising host tree and fungal volatiles were detected and classified into different groups, namely, aliphatic hydrocarbons (17 compounds), aromatics (2 compounds), monoterpene hydrocarbons (15 compounds), oxygenated monoterpenes (26 compounds), sesquiterpenes (17 compounds), and spiroketals (3 compounds). The proportion of total oxygenated monoterpenes gradually increased to dominate the volatile profile of *G. penicillata*-infected bark reaching a maximum at 18 d post-inoculation during our sampling period ($F_{(3,14)} = 5.51$, $p = 0.01$, ANOVA), while the proportion of total oxygenated monoterpenes was unchanged in mock-inoculated controls (Fig 2B). Camphor was the major contributor to the overall increase of oxygenated monoterpenes, with the highest relative abundance after 18 d and a significant difference between time points ($F_{(1,16)} = 13.06$, $p = 0.002$, ANOVA) (S5 Table). The proportion of monoterpene

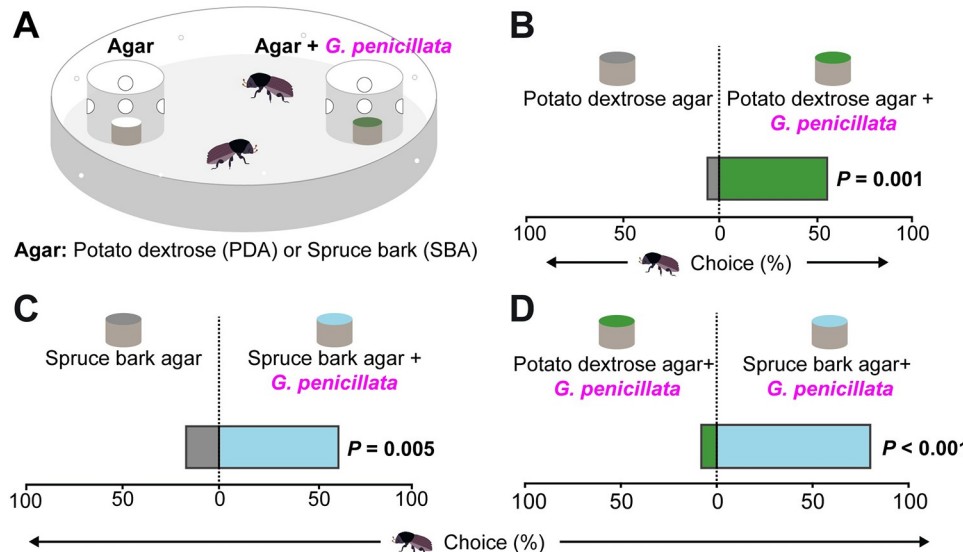

**Fig 1. *Ips typographus* uses volatile cues from spruce bark inoculated with *G. penicillata (Gp)* to detect this symbiotic fungus. (A)** Arena used for trap bioassays to study the behavior of *I. typographus* to volatiles emitted by a symbiotic fungus. Cups containing agar (with and without fungus) were offered to beetles as odor samples. Holes on each side of the cup allowed the beetles to smell but not touch the agar until they entered the cup and then could no longer escape. **(B, C)** Adult beetles chose *Gp*-colonized agar medium over fungus-free medium (*n* = 25, with two beetles per replicate). **(D)** Adult beetles chose *Gp*-inoculated spruce bark agar over *Gp*-inoculated agar without spruce bark (*n* = 25). **(B–D)** Deviation of response indices against zero was tested using Wilcoxon's test. The data underlying this Figure can be found at https://doi.org/10.6084/m9.figshare.21692156.v1.

hydrocarbons gradually decreased over time in both treatments (control, $F_{(3,7)}$ = 11.6, $p$ = 0.004; *G. penicillata*, $F_{(3,14)}$ = 11.5, $p$ < 0.001, ANOVA). The proportion of total sesquiterpenes also decreased significantly over time in *G. penicillata*-infected bark ($F_{(3,15)}$ = 4.4, $p$ = 0.02, ANOVA), but not in mock-inoculated control bark. When measuring emission rate (ng mg $DW^{-1}$ $h^{-1}$), the emission of total monoterpene hydrocarbons in the control and the *G. penicillata*-treated bark plugs was not significantly different at 4 d post-inoculation (Fig 2C and S2 Table). However, there was a dramatic increase in the emission rate of total oxygenated monoterpenes at this time point in spruce bark inoculated with *G. penicillata* compared to the mock-inoculated control (Fig 2D) (9-fold increase, t(3.2) = 7.38, $p$ = 0.004, Welch's *t* test). Nineteen oxygenated monoterpenes were identified in the bark inoculated with *G. penicillata*, and a total of 15 compounds significantly increased in fungus infected bark compared to the control (Fig 2E) (S2 Table) including camphor (51-fold increase, t(3) = 7.7, $p$ = 0.004, Welch's *t* test), *endo*-borneol (18-fold increase, t(6) = 6.5, $p$ < 0.001), isopinocamphone (3-fold increase, t(6) = 6.8, $p$ < 0.001), verbenone (2-fold increase, t(6) = 4.3, $p$ = 0.005), and bornyl acetate (3-fold increase, t(6) = 3.7, $p$ = 0.01). Measurements conducted on four other *I. typographus* fungal symbionts also showed differences in the volatile composition of fungus-inoculated versus control bark (S2 Fig), with increases in the proportion of oxygenated monoterpenes in bark inoculated with *L. europhioides* and *O. bicolor* over time, but not for *E. polonica* (S3 Fig and S4, S6, and S7 Tables).

## Symbiotic fungi convert spruce monoterpenes into oxygenated products

The symbiotic fungus, *G. penicillata*, significantly decreased the amount of (−)-bornyl acetate added to PDA medium after 4 d compared to a fungus-free control (Fig 3A) (*t* = −3.38,

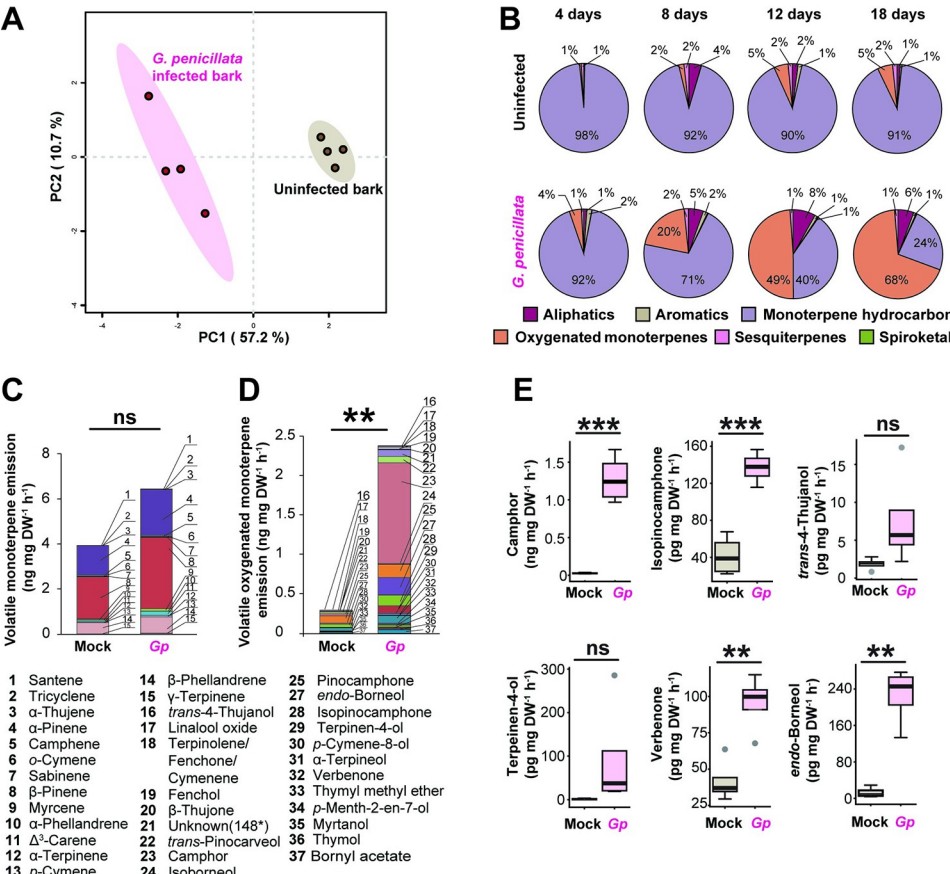

**Fig 2. Growth of *Ips typographus* symbiotic fungi on spruce bark induces increased emission of oxygenated monoterpenes.** **(A)** Volatile emission pattern differed between spruce bark inoculated with *G. penicillata* (*Gp*) and uninfected bark 4 days after inoculation, as depicted in a partial least squares discriminant analysis (PLS-DA). Complete volatile emission data by compound for *G. penicillata* and other *I. typographus* fungal symbionts are given in S2 Table. **(B)** Changes in volatile emission profiles of spruce bark due to *G. penicillata* infection over an 18-d time course. Compounds are classified into six groups by their chemical structures (*n* = 3 to 5). Each circle displays the relative amounts of each class of compounds at a single time point and does not indicate changes in the total amounts of volatiles. Complete volatile emission data by compound and time point for *G. penicillata* and other symbionts are given in S3–S6 Tables. **(C, D)** Emission of specific monoterpenes from fresh spruce bark inoculated with *G. penicillata* at 4 days post-inoculation. Identified compounds were classified into monoterpene hydrocarbons **(C)** and oxygenated monoterpenes **(D).** The individual compounds are stacked within a single bar representing the total emission. Significant differences in the total emission levels induced by *G. penicillata* are denoted by asterisks above the bars (*n* = 4, Welch's *t* test with ns = not significant $^*P < 0.05$, $^{**}P < 0.01$, $^{***}P < 0.001$). The numbers denote the identities of the compounds in the stacked bars. Complete volatile emission data are given in S2 Table. **(E)** Emission rate of major oxygenated monoterpenes from spruce bark inoculated with *G. penicillata* 4 d post-inoculation. Asterisks indicate significant differences between the spruce bark-inoculated *G. penicillata* and the fungus-free control (Welch's *t* test). The data underlying this Figure can be found at https://doi.org/10.6084/m9.figshare.21692156.v1.

*p* = 0.003). However, the amounts of two other monoterpenes, (−)-β-pinene and (−)-α-pinene, did not differ between *G. penicillata*-inoculated and fungus-free PDA.

The monoterpene metabolites of *G. penicillata* grown on either (−)-α-pinene- or (−)-β-pinene-enriched agar were similar, with the oxygenated monoterpene terpinen-4-ol being the major product from both spruce precursors (Fig 3B). Other oxygenated monoterpenes formed by *G. penicillata* in the presence of (−)-α-pinene and (−)-β-pinene included (+)-isopinocamphone, (+)-*trans*-4-thujanol and α-terpineol, with (−)-verbenone being detected only from (−)-α-pinene (S5 and S6 Figs). From (−)-bornyl acetate, the dominant oxygenated

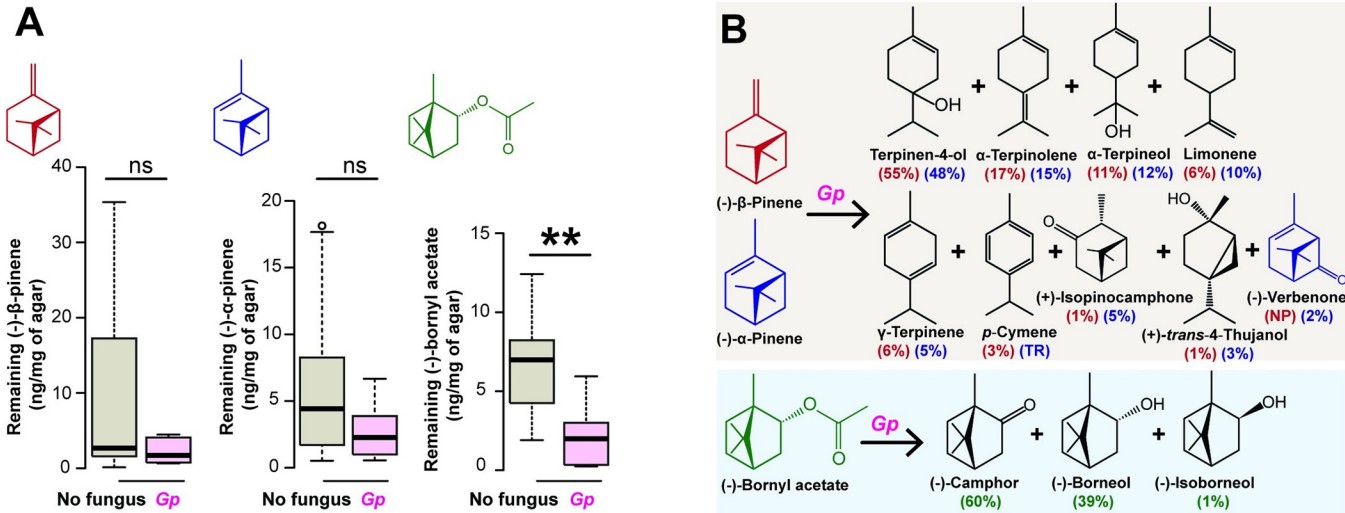

**Fig 3. Metabolism of major spruce monoterpenes by a symbiotic fungus produces a variety of oxygenated monoterpenes. (A, B)** Metabolism of major spruce monoterpenes by *G. penicillata* after fungal-infected vs. uninfected PDA was supplemented with 0.5 mg/g of (−)-β-pinene, (−)-α-pinene, and (−)-bornyl acetate. **(A)** The amounts of starting monoterpenes remaining after 4 days. Error bars represent SEM (*n* = 5 to 13). Asterisks indicate a significant difference between the PDA-inoculated *G. penicillata* and the fungus-free control (Welch's *t* test) with ns = not significant, ** $P < 0.01$. **(B)** The most abundant metabolites of administered monoterpenes are depicted with their percentages relative to the total amounts of metabolites detected for each compound (derived from (−)-β-pinene in red; derived from (−)-α-pinene in purple; derived from (−)-bornyl acetate in green). Amounts were determined by headspace collection of volatiles from the agar (*n* = 3). The data underlying this Figure can be found at https://doi.org/10.6084/m9.figshare.21692156.v1.

monoterpenes of *G. penicillata* were camphor and *endo*-borneol (Fig 3B, bottom panel; S8 Fig), and their production coincided with the decrease of the precursor (Fig 3A). Similar results were obtained for the three other fungal symbionts tested (S4–S8 Figs).

The volatile blend of the co-occurring, but nonbeneficial fungus, *Trichoderma* sp., produced from a synthetic spruce monoterpene mixture (S8 Table) was distinct from that of *G. penicillata* (S9 Fig). *Trichoderma* sp. produced mainly the oxygenated monoterpenes, (−)-borneol and (−)-verbenone, while *G. penicillata* produced mainly (−)-camphor and terpinen-4-ol. These results collectively show that symbiotic fungi can alter the volatile profile of spruce bark by increasing the emission of oxygenated monoterpenes, and the emission profile of symbiotic fungi is distinct from that of a co-occurring fungal saprophyte.

### Elimination of symbiotic ophiostomatoid fungi from *I. typographus* reduces the production of oxygenated monoterpenes in bark beetle galleries

To determine the primary source of oxygenated monoterpenes in beetle galleries, unsterilized pupae were reared into fungus-free mature adults using a diet fortified with the fungistatic agent sodium benzoate (S1 Method). Sterility tests and scanning electron microscopy of beetle body parts revealed that beetles reared on this diet were free from fungi and the elytral pits were devoid of fungal spores and yeasts (S15C Fig and S9 Table), although bacteria were abundant. In comparison, elytral pits of beetles reared on diet without sodium benzoate contained abundant spore masses, with structural similarity to those of *O. bicolor* and yeasts (S15B Fig and S9 Table) [40]. Reinoculation of fungus-free beetles with *G. penicillata* resulted in acquisition of *G. penicillata* spores in the elytral pits (S15D Fig).

Next, we quantified the amount of oxygenated monoterpenes present in the bark galleries constructed by untreated, fungus-free, and *G. penicillata*-reinoculated beetles separately using

GC–MS analyses of gallery extracts. The concentration of (−)-camphor was significantly higher in the galleries of reinoculated beetles than in bark samples without beetles at all ($F$ ($3,23$) = 15.6, $p < 0.001$, ANOVA), while the concentration of terpinen-4-ol was significantly higher in untreated beetles than in both fungus-free beetles or in bark samples without beetles ($H$ ($3$) = 18.6, $p < 0.001$, Kruskal–Wallis) (Fig 4B). On the other hand, the concentrations of these two oxygenated monoterpenes were similar when comparing galleries of *G. penicillata*-reinoculated beetles to those of untreated beetles. For (+)-isopinocamphone, the concentration in galleries constructed by untreated beetles was significantly higher compared to bark samples without beetle galleries, but similar to that in galleries constructed by fungus-free and *G. penicillata*-reinoculated beetles ($F$ ($3,23$) = 5.7, $p = 0.005$, ANOVA) (Fig 4B). On the other hand, the concentration of (−)-borneol in fungus-free galleries was significantly higher compared to that in bark samples without galleries and was similar to that in the galleries of *G. penicillata*-reinoculated and untreated beetles ($F$ ($3,23$) = 5.3, $p = 0.006$, ANOVA) (Fig 4B). There were no significant differences in the concentrations of pinocamphone, α-terpineol, and (−)-verbenone among treatments.

Analysis of the microbes present in each treatment revealed that galleries constructed by fungus-free beetles were indeed free from symbiotic ophiostomatoid fungi and yeasts but contained many saprophytes and (or) endophytes and bacteria (S10 Table). Microbes present in the galleries of untreated and *G. penicillata*-reinoculated beetles were similar to those found on beetles that were used to infest the bark (S9 and S10 Tables).

## *Ips typographus* detects oxygenated monoterpenes through specialized olfactory sensory neurons (OSNs) in their antennae

To test if *I. typographus* antennal olfactory sensilla contain OSNs that detect the biotransformation products of monoterpenes, we challenged 231 olfactory sensilla with a test panel comprising 92 ecologically relevant compounds diluted in paraffin oil (1 µg µl$^{-1}$) using single cell recordings (S1 Table). Only 23 (approximately 10%) of the sensilla housed neurons that did not respond to any of the compounds from the odor panel, although their OSNs showed spontaneous firing. We obtained odor-evoked responses with strong excitation (>120 Hz) from 198 OSNs and weak excitation (<50 Hz) from 10 OSNs, allowing the grouping of these neurons into different classes based on their response profiles. From initial screening experiments at a 10-µg dose on filter paper (to determine the maximum receptive range of OSNs), we identified 20 classes of OSNs. Three OSN classes responded primarily to fungal-produced oxygenated monoterpenes. We also identified neurons belonging to previously described OSN classes tuned to pheromones, host tree volatiles, and nonhost odorants [41] that are not further considered here.

OSN classes tuned to fungal-produced oxygenated monoterpenes were identified in both the $A_m$ and $B_m$ regions on the antennae (Fig 5A). One of these OSN classes responded most strongly to (+)-isopinocamphone, and this class was highly specific for oxygenated monoterpenes, especially ketones (Fig 5C, left panel). Apart from (+)-isopinocamphone, relatively strong responses were also elicited by (+)-pinocamphone, (−)-isopinocamphone, (±)-pinocarvone, (±)-camphor, and (−)-pinocamphone (Fig 5C, left panel). Dose–response tests showed that this OSN class was the most sensitive to (+)-isopinocamphone of all the compounds tested with responses evident at a dose of 100 pg. The responses to (+)-pinocamphone, (−)-isopinocamphone, (±)-pinocarvone, and (±)-camphor all appeared between 1 ng and 10 ng doses (Fig 5D, left panel). Another OSN class with specific responses to fungal-derived compounds responded most strongly to (+)-*trans*-4-thujanol and weakly to (±)-3-octanol, (±)-1-octen-3-ol, (+)- and (−)-terpinen-4-ol, and (+)- and (−)-α-terpineol (Fig 5C, middle panel). This

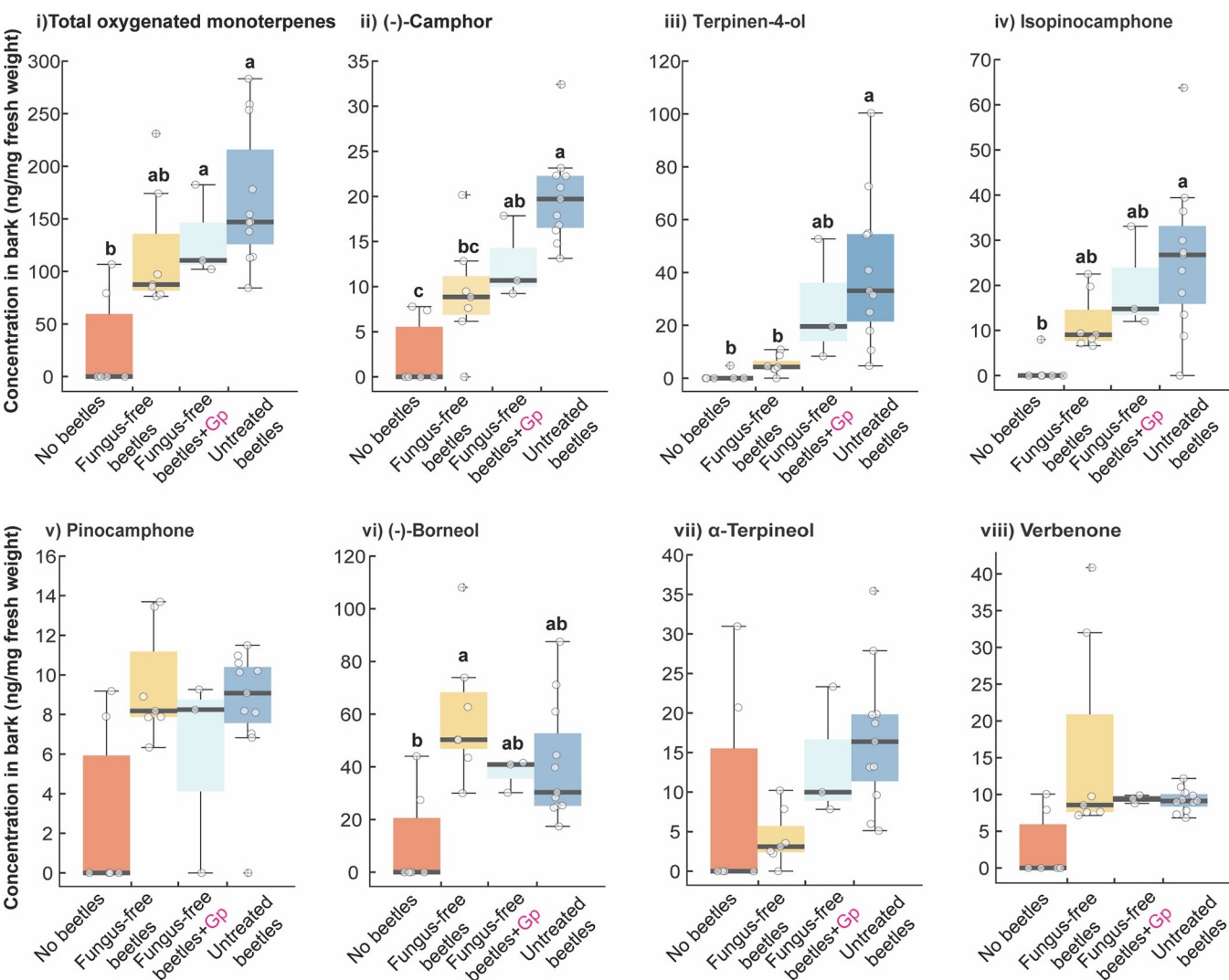

**Fig 4. Oxygenated monoterpenes identified in *I. typographus*-infested spruce bark are produced by fungi associated with bark beetles.** Concentration of total oxygenated monoterpenes (i), camphor (ii), terpinen-4-ol (iii), isopinocamphone (iv), pinocamphone (v), borneol (vi), α-terpineol (vii), and verbenone (viii) in bark without beetles, infested with fungus-free beetles, infested with fungus-free beetles reinoculated with *G. penicillata*, and untreated beetles. Different lowercase letters indicate significant differences between treatments (ANOVA, Tukey's test, $P < 0.05$). The data underlying this Figure can be found at https://doi.org/10.6084/m9.figshare.21692156.v1.

OSN showed a 1,000-fold lower response threshold to (+)-*trans*-4-thujanol compared to the next best ligands, the $C_8$ alcohols (Fig 5D, middle panel). Finally, an OSN class responding strongly to verbenone, α-isophorone and β-isophorone, followed by weaker responses to (−)- and (+)-*trans*-verbenol, pinocarvone, and (−)-*cis*-verbenol (Fig 5C, right panel) was also found. Dose–response tests revealed that this neuron class responded the strongest to α-isophorone across most tested doses, followed by slightly weaker and similarly strong responses to both verbenone and β-isophorone (Fig 5D, right panel).

A few previously characterized OSN classes for host tree monoterpenes, including the classes with primary responses to α-pinene, *p*-cymene, and Δ3-carene, respectively [41], showed varying secondary responses to some of the fungal-derived compounds tested here for the first time. For example, the α-pinene OSN class responded also to

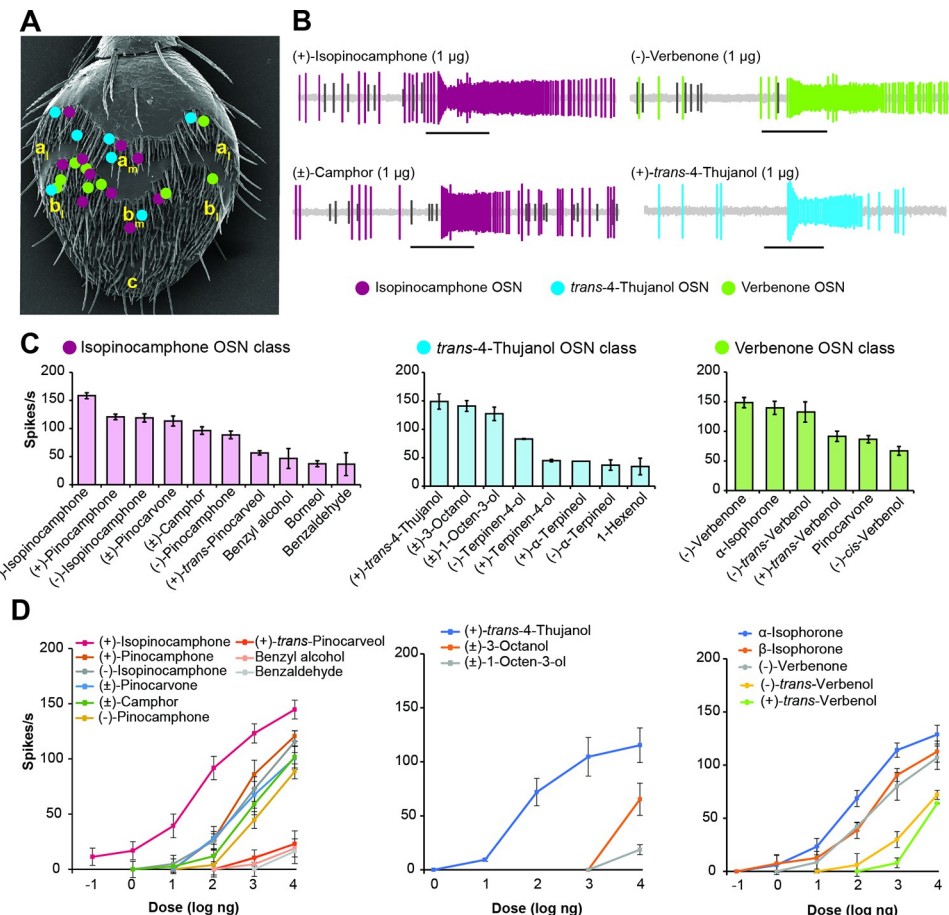

**Fig 5. Oxygenated monoterpenes derived from fungal metabolism of host tree monoterpene hydrocarbons are detected by specialized olfactory neurons in the *I. typographus* antenna.** (A) Mapping of three classes of olfactory sensory neurons (OSN) selective for oxygenated monoterpenes (isopinocamphone; verbenone; (−)-*trans*-4-thujanol) on the antenna. Olfactory sensilla housing these OSN classes are distributed in medial (subscript "m") and lateral (subscript "l") regions of sensillum areas a and b (but not in area c) of the antenna. (B) Representative spike trains from an isopinocamphone-responsive neuron stimulated with 1 μg of (+)-isopinocamphone (top left) and (±)-camphor (bottom left); a verbenone-responsive neuron stimulated with 1 μg (−)-verbenone (top right); a (+)-*trans*-4-thujanol-responsive neuron stimulated with 1 μg (+)-*trans*-4-thujanol (bottom right). Black horizontal bars indicate the 0.5-s odor puffs. (C) Response spectra of OSN classes responding predominantly to oxygenated monoterpenes produced by fungi at the 10-μg screening dose. The average number of spikes/second was recorded from the isopinocamphone-tuned OSN class (left) ($n = 6$ except (+) and (−)-pinocamphone ($n = 3$)), *trans*-4-thujanol-tuned OSN class (middle) ($n = 5$), and verbenone and isophorone-tuned OSN class (right) ($n = 4$) after neurons were stimulated with a panel of 97 odors. Error bars represent SEM. (D) Dose–response curves of the OSNs stimulated with their most active ligands: isopinocamphone-tuned OSN class (left) (($n = 9$) except for (+) and (−)-pinocamphone ($n = 3$)), *trans*-4-thujanol-tuned OSN class (middle) ($n = 3$), and the OSN class tuned to isophorone and verbenone (right) ($n = 3$). Error bars represent SEM. The data underlying this Figure can be found at https://doi.org/10.6084/m9.figshare.21692156.v1.

(+)-isopinocamphone, (−)-isopinocamphone, and (±)-pinocarvone and weakly to (±)-camphor, (−)-myrtenol, *trans*-pinocarveol, carvone, borneol, and (−)-fenchone (S10A Fig). The *p*-cymene OSN class showed intermediate responses to (+)-*trans*-4-thujanol and carvone (S10B Fig). Although the Δ3-carene OSN class showed high specificity towards Δ3-carene, ligands such as camphor and (−)-isopinocamphone also elicited weak responses from this neuron class (S10C Fig).

## Oxygenated monoterpenes produced by fungal symbionts attract or repel *I. typographus*

*G. penicillata* growing on SBA with added 0.1 mg g$^{-1}$ or 0.5 mg g$^{-1}$ (−)-β-pinene (within the range of natural concentrations in *P. abies* bark [42,43] was much more attractive to adult *I. typographus* than a fungus-free control medium supplemented with (−)-β-pinene (Fig 6A, left) (0.1 mg g$^{-1}$, $z = 2.22$, $p = 0.02$; 0.5 mg g$^{-1}$, $z = 2.54$, $p = 0.01$, Wilcoxon's test). This attraction was correlated with the presence of oxygenated monoterpenes produced by the fungus from (−)-β-pinene (Fig 6A, right). However, supplementation of agar with 1 mg g$^{-1}$ (−)-β-pinene did not lead to a significant choice between fungus and fungus-free (−)-β-pinene-containing medium (Fig 6A, left). To further understand this concentration-dependent response, individual (−)-β-pinene biotransformation products that had been shown to be electrophysiologically active (see last section) were used in trap bioassays at various doses against a mineral oil control. Adult beetles were significantly attracted to 100 μg *trans*-4-thujanol (Fig 6B, left panel) ($z = 2.78$, $p = 0.005$, Wilcoxon's test), but other concentrations of *trans*-4-thujanol and all concentrations of (+)-isopinocamphone and terpinen-4-ol tested had no significant effect. Adult beetles did not discriminate between *G. penicillata* grown on (−)-β-pinene enriched medium and *G. penicillata* grown on nonenriched medium (Fig 6E). Based on these results, we conclude that *I. typographus* shows concentration-specific responses to some oxygenated fungal metabolites of (−)-β-pinene, but not to (−)-β-pinene itself (S11 Fig).

Addition of another host tree monoterpene, (−)-bornyl acetate, to fungal growth medium at 0.05 mg g$^{-1}$ (in the range of natural concentrations in *P. abies* bark) and 0.5 mg g$^{-1}$ resulted in strong attraction of *I. typographus* adults towards *G. penicillata* when tested against a fungus-free control after 4-d incubation (Fig 6C, left panel) (0.05 mg g$^{-1}$, $z = 3.31$, $p = 0.001$; 0.5 mg g$^{-1}$, $z = 3.21$, $p = 0.001$, Wilcoxon's test). This attraction was correlated with the presence of fungal biotransformation products from (−)-bornyl acetate (Fig 6C, right panel). The major biotransformation product that formed in this period (Fig 6C) was camphor, which was significantly more attractive to adult beetles at a 100-μg dose than the mineral oil control (Fig 6D) ($z = 2.58$, $p = 0.01$, Wilcoxon's test). However, camphor had no effect at larger or smaller doses. Adult beetles preferred *G. penicillata* grown on unenriched medium against *G. penicillata* grown on medium enriched with a high amount of (−)-bornyl acetate (0.5 mg g$^{-1}$) (Fig 6F) ($z = 2.12$, $p = 0.03$, Wilcoxon's test). By contrast, in the absence of fungus, beetles did not discriminate between diet enriched with monoterpenes and diet without monoterpenes (S11 Fig). Thus, *I. typographus* is attracted to the major oxygenated fungal metabolite of (−)-bornyl acetate but is not attracted and even repelled by (−)-bornyl acetate itself.

## Volatiles of symbiotic fungi increase *I. typographus* attraction to pheromones

Female beetles were significantly more attracted towards the individual aggregation pheromone components *cis*-verbenol and 2-methyl-3-buten-2-ol (*cis*-verbenol, $z = 2.98$, $p = 0.003$; 2-methyl-3-buten-2-ol, $z = 2$, $p = 0.046$, Wilcoxon's test), and towards a mixture of the two pheromone components ($z = 5.19$, $p < 0.001$, Wilcoxon's test) compared to the mineral oil control (Fig 7A). By contrast, adult males did not make a choice between these options in accordance with previous studies [43,44]. However, when beetles had to choose between the pheromone mixture with or without *G. penicillata* volatiles, females preferred the pheromone mixture together with *G. penicillata* volatiles (Fig 7B) ($z = 3.41$, $p = 0.001$, Wilcoxon's test).

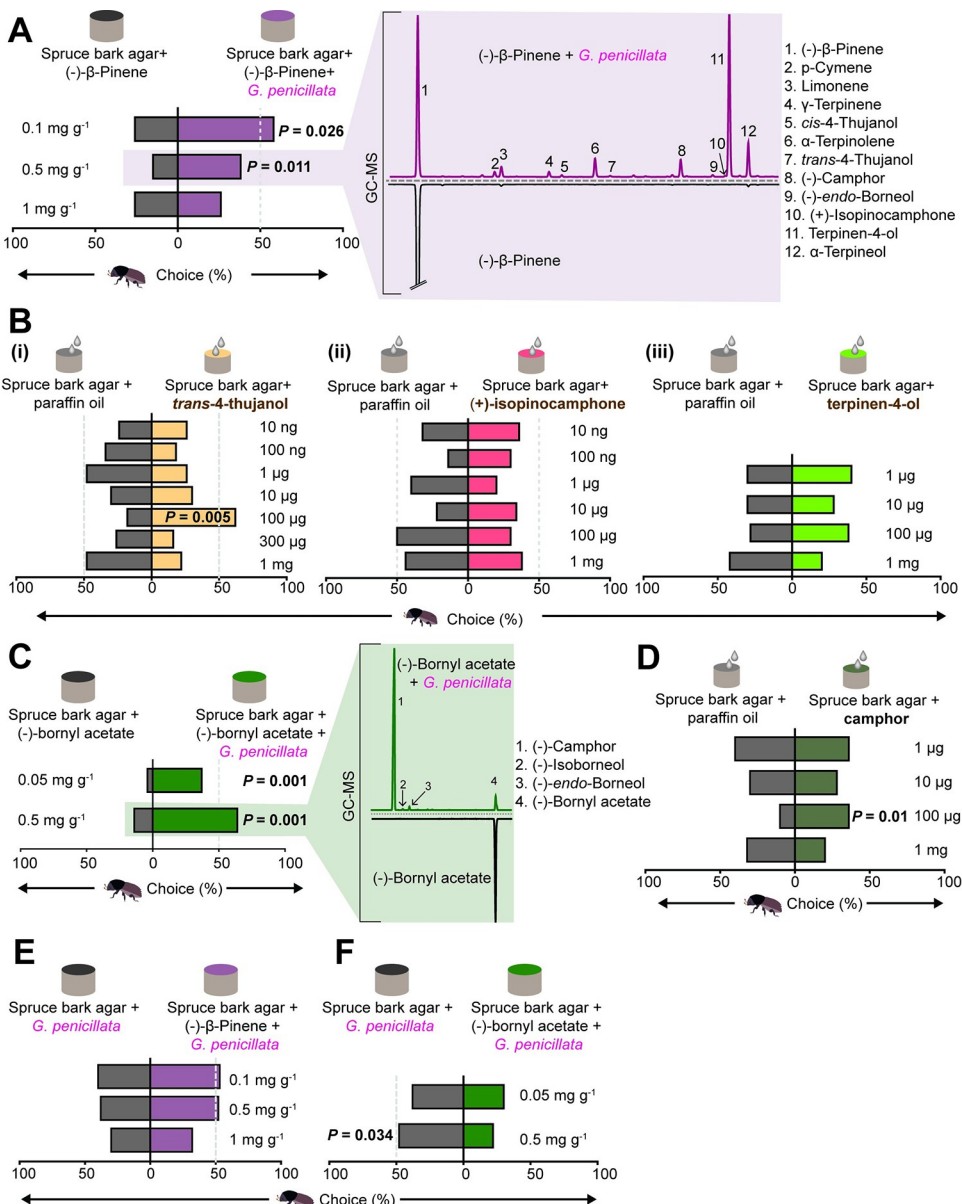

**Fig 6. *Ips typographus* is attracted to a fungal symbiont that produces a blend of oxygenated monoterpenes. (A)** Adult beetles preferred spruce bark agar enriched with 0.1 and 0.5 mg/g of (−)-β-pinene and inoculated with *G. penicillata* over spruce bark agar enriched with 0.1 and 0.5 mg/g (−)-β-pinene alone, but without fungus (left). GC–MS traces depict the headspace volatiles of (−)-β-pinene-enriched agar left for 4 d with and without *G. penicillata* (right) and show several oxygenated monoterpenes produced by the fungus from (−)-β-pinene. Numbers refer to the identities of compounds. **(B)** Adult beetles chose *trans*-4-thujanol (left) at a 100-μg dose diluted in mineral oil, when tested against a mineral oil control (Bonferroni adjusted significance level, α = 0.007). Adult beetles showed indifferent responses to (+)-isopinocamphone (middle), and terpinen-4-ol (right), applied in various doses in mineral oil. **(C)** Adult beetles preferred spruce bark agar enriched with various amounts of (−)-bornyl acetate and inoculated with *G. penicillata* over spruce bark agar enriched with (−)-bornyl acetate but without fungus (left). GC–MS traces depict the headspace volatiles of 0.5 mg/g (−)-bornyl acetate-enriched agar left for 4 d with and without *G. penicillata* (right) and show several oxygenated monoterpenes produced by the fungus from (−)-bornyl acetate. Numbers refer to the identities of compounds. **(D)** Adult beetles preferred (±)-camphor at a 100-μg dose against a mineral oil control, but not at other doses (Bonferroni adjusted α = 0.012). **(E)** Adult beetles did not discriminate between *G. penicillata* on agar with three different concentrations of (−)-β-pinene and *G. penicillata* without (−)-β-pinene. **(F)** Adult beetles chose *G. penicillata* on agar without any administrated (−)-bornyl acetate vs. *G. penicillata* on agar with 0.5 mg/g (−)-bornyl acetate. **(A–F)** Deviation of response indices against zero was tested using Wilcoxon's test. *n* = 25 or 50 for each trial. The data underlying this Figure can be found at https://doi.org/10.6084/m9.figshare.21692156.v1.

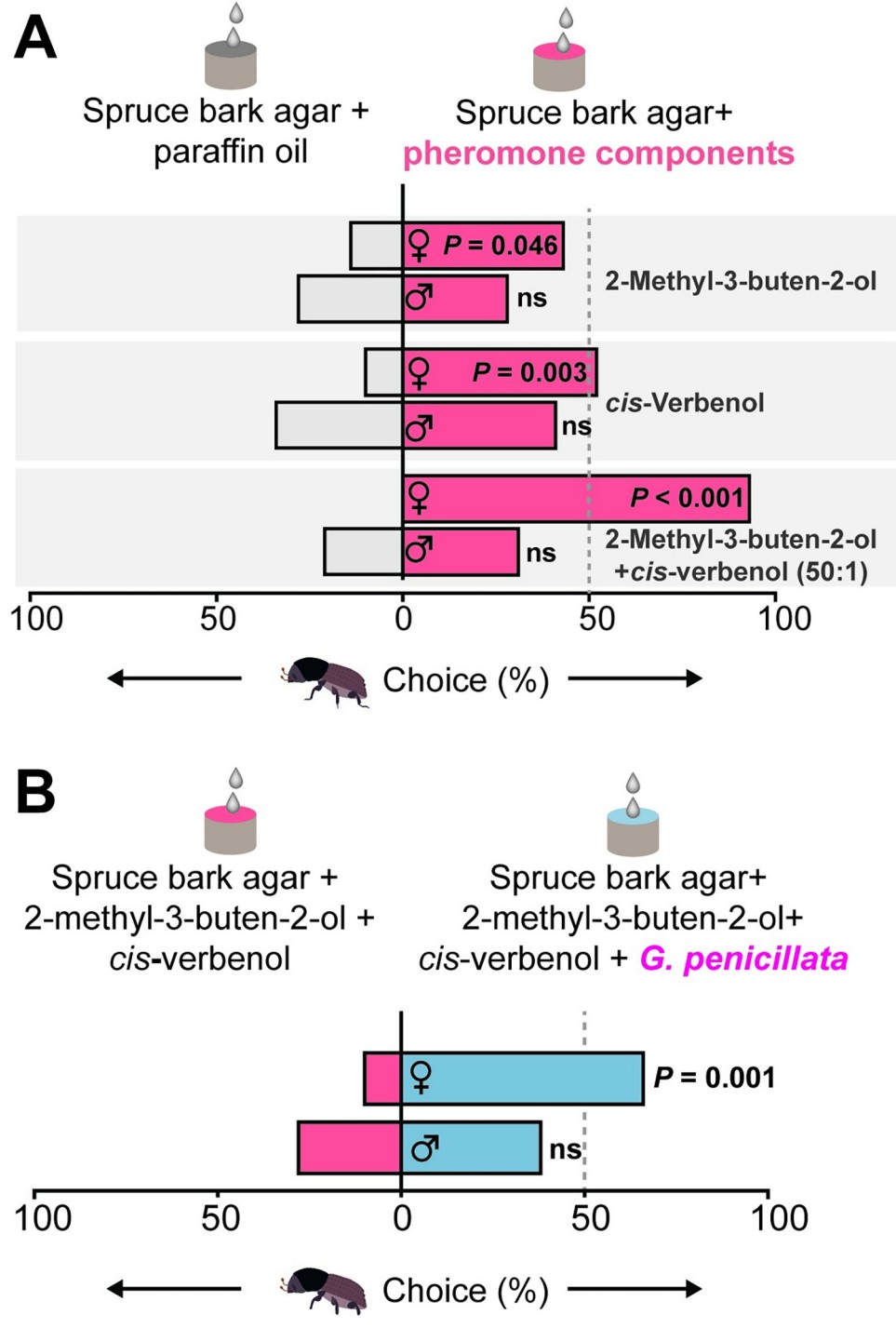

**Fig 7. Female adult *I. typographus* are attracted towards a pheromone mixture in the presence of *G. penicillata* volatiles. (A)** Adult females chose traps containing 2-methyl-3-buten-2-ol and *cis*-verbenol at $10^{-2}$ concentration diluted in mineral oil over control traps containing mineral oil. Females strongly preferred traps containing a binary pheromone blend (*cis*-verbenol: 2-methyl-3-buten-2-ol, 50:1) diluted in mineral oil over the mineral oil control (bottom). Adult males were unresponsive to these concentrations of individual pheromones and their blend. **(B)** Adult females preferred pheromone blend in the presence of *G. penicillata* volatiles over pheromone blend without fungus. **(A, B)**. Deviation of response indices against zero was tested using Wilcoxon's test ($n = 28$ for each experiment). The data underlying this Figure can be found at https://doi.org/10.6084/m9.figshare.21692156.v1.

### Symbiotic fungi increase the tunneling of adult *I. typographus*

The presence of a symbiotic fungus increased the tunneling of adult *I. typographus* beetles in medium after 48 h. Multiple logistic regression analysis revealed that successful tunneling odds in (−)-bornyl acetate-amended medium were significantly influenced by the fungus when the beetle sex and the monoterpene treatment remained constant (Table 1) ($\beta$ = 4.98, $\chi^2$ = 20.99, $p < 0.001$). The presence of the fungus increased the tunneling probability by 99% compared to in the absence of the fungus. Additionally, males had significantly lower tunneling success compared to females (Table 1) ($\beta$ = −1.72, $\chi^2$ = 8.78, $p = 0.003$) with the tunneling probability for males being 15.2% lower than for females. In (−)-β-pinene-amended medium, only the presence of the fungus significantly influenced the tunneling probability of beetles (Table 1) ($\beta$ = 2.65, $\chi^2$ = 20.45, $p < 0.001$), with an increase of 93% compared to in the absence of the fungus. Similarly, in (−)-α-pinene-amended medium, only the presence of the fungus significantly influenced the tunneling probability of beetles (Table 1) ($\beta$ = 2.02, $\chi^2$ = 9.39, $p = 0.002$), with an increase of 88% compared to the absence of the fungus. Addition of the three monoterpenes without fungus did not have any effect on the tunneling behavior of adult beetles.

The growth of *G. penicillata* on monoterpene-enriched media resulted in significantly longer beetle tunnels than in the other treatments (Fig 8B–8D) ((−)-β-pinene, $F_{(3,99)}$ = 4.95, $p = 0.003$; (−)-α-pinene, $F_{(3,92)}$ = 14.8, $p < 0.001$; (−)-bornyl acetate, $F_{(3,106)}$ = 6.6, $p < 0.001$, ANOVA, Tukey's test)). However, there were no other significant interactions between monoterpenes and *G. penicillata* affecting tunneling, i.e., no other differences in tunnel lengths when comparing fungus alone versus fungus plus monoterpenes except for the treatment with (−)-β-pinene (Fig 8B–8D, Tukey's test). The sex of the beetle and the interaction of sex with other treatments also had no effect on the tunnel length.

## Discussion

The successful attack of bark beetles on their host trees is invariably associated with free-living fungal symbionts. These symbionts may detoxify the terpene-rich defensive resin of the host tree, hasten host tree death, provide nutritional benefits, and increase resistance to pathogens [16,17,21]. Here, we document the ability of *I. typographus* fungal symbionts to metabolize host tree monoterpenes to oxygenated derivatives based on experiments in which we cultured fungi with and without host tree monoterpenes. Several oxygenated monoterpenes have been previously identified as volatiles released from trees that were attacked by *I. typographus* [32,33,35]. We showed that these are fungal metabolites that become dominant components of the volatile profile making up over half of the total bark volatiles emitted 12 d following infection. Bark beetle OSNs can detect many of these oxygenated monoterpenes, and some of these compounds attract adult beetles to diets inoculated with fungal symbionts. We hypothesize that oxygenated monoterpenes may assist adult beetles in locating suitable breeding and feeding sites [34,45].

### Bark beetle symbiotic fungi metabolize host tree monoterpenes to oxidized derivatives

Fungi are well known to metabolize monoterpenes as well as many other chemical compounds they encounter in their natural environments [46,47]. In particular, fungi associated with conifer trees are reported to transform different types of resin monoterpenes to oxidized derivatives [48,49]. Previous literature describes this capability for a symbiotic fungus of the North American bark beetle *Dendroctonus ponderosae* [50]. Here, we showed that fungal symbionts

**Table 1. Multiple logistic regression analysis predicting the odds of adult _I. typographus_ tunneling into media enriched in different monoterpenes with and without _G. penicillata_ in a no-choice assay (see also Fig 6A).** The data underlying this Table can be found at https://doi.org/10.6084/m9.figshare.21692156.v1.

| Successful tunneling in diets with[a] | Predictors | β | SE[b] | Wald χ² | _P_ value | Exp(β) | 95% CI for Exp(β) | |
|---|---|---|---|---|---|---|---|---|
| | | | | | | | Lower | Upper |
| (−)-Bornyl acetate | Monoterpene | −0.31 | 0.56 | 0.31 | ns | 0.73 | 0.25 | 2.19 |
| | Sex | −1.72 | 0.58 | 8.78 | **0.003** | 0.18 | 0.06 | 0.56 |
| | Fungus | 4.98 | 1.09 | 20.99 | **<0.001** | 145.30 | 17.27 | 1222.52 |
| (−)-β-Pinene | Monoterpene | −0.87 | 0.54 | 2.57 | ns | 0.42 | 0.14 | 1.22 |
| | Sex | 0.25 | 0.52 | 0.24 | ns | 1.29 | 0.47 | 3.55 |
| | Fungus | 2.65 | 0.58 | 20.45 | **<0.001** | 14.09 | 4.48 | 44.36 |
| (−)-α-Pinene | Monoterpene | 0.00 | 0.52 | 0.00 | ns | 1.00 | 0.36 | 2.75 |
| | Sex | 0.00 | 0.52 | 0.00 | ns | 1.00 | 0.36 | 2.75 |
| | Fungus | 2.02 | 0.66 | 9.39 | **0.002** | 7.51 | 2.07 | 27.28 |

[a]The reference category is unsuccessful tunneling.

[b]Standard error of _β_.

ns, not significant.

of the European spruce bark beetle _I. typographus_ also oxidize monoterpenes of their host trees by adding monoterpene standards to the culture medium and by growing symbionts on spruce bark with natural levels of monoterpenes.

Oxidation of monoterpenes can be considered a detoxification process since these lipophilic compounds have antifungal properties [49,50], and oxidation may facilitate excretion by increasing polarity. However, individual oxidative transformations may not necessarily decrease toxicity but could serve as entry points to reaction cascades that allow monoterpenes to be degraded and used as carbon sources for intermediary metabolism [50,51]. Here, we only searched for the volatile monoterpene metabolites of fungal symbionts and so cannot speculate on whether or not monoterpenes are employed as carbon sources.

Although we did not use isotopically labeled substances to confirm that fungal symbionts metabolize host tree monoterpenes to oxygenated derivatives, the de novo synthesis of oxygenated cyclic monoterpenes by these fungi can be almost ruled out. In previous work, we collected volatiles from _G. penicillata_ and the other _I. typographus_-associated fungi studied here growing under many conditions without spruce bark or supplemental monoterpenes [38]. While we detected a large range of volatile compounds, we never found any trace of oxidized cyclic monoterpenes like those reported here, which all have the bornane, _p_-menthane, pinane, or thujane carbon skeletons characteristic of spruce monoterpenes. Some fungi do produce cyclic monoterpenes of their own, but none have yet been reported to synthesize the variety of cyclic skeletons produced by conifers, and so are very likely to obtain their precursors for oxygenated monoterpenes from conifer trees.

### Oxygenated monoterpenes are widespread volatile cues for tree-feeding insects

Various forest insects that are associated with fungal symbionts, such as bark beetles, ambrosia beetles, and wood wasps, live in host trees producing large quantities of monoterpene volatiles. These insects are often attracted to their fungal symbionts through volatiles [2,38,52], and, hence, fungus-produced monoterpene metabolites could be critical components of the attractive volatile blends. For bark beetles, oxygenated monoterpenes derived from either the beetles, fungi, or host trees are already known to play important roles in their life history [53]. During

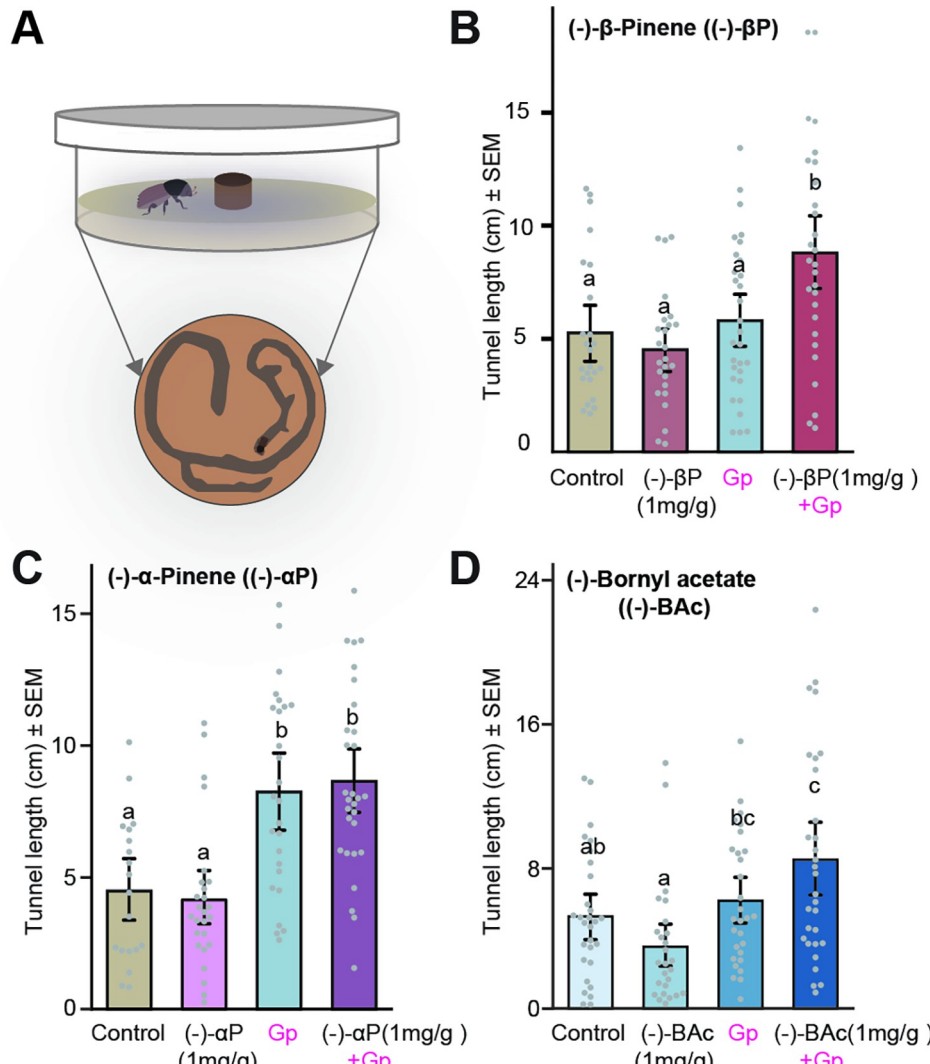

**Fig 8.** *Ips typographus* **tunnels more in monoterpene-enriched diet in the presence of a symbiotic fungus. (A)** Schematic drawing of a Petri dish arena used for the no-choice tunneling assay. The dish was filled with monoterpene-enriched spruce bark agar and inoculated with *G. penicillata* (Gp) (top). Example of the tunneling pattern of an adult beetle within the fungus-colonized diet as pictured from the bottom side of Petri dish (bottom). **(B–D)** Tunnel lengths (cm) made by adult beetles after 48 h in diet containing *G. penicillata* only, monoterpenes only, *G. penicillata* and monoterpenes, or controls with neither *G. penicillata* nor monoterpenes. Error bars represent SEM ($n = 30$ (15 ♂, 15 ♀) for each trial). Monoterpenes: (−)-β-pinene **(B)**, (−)-α-pinene **(C)**, (−)-bornyl acetate **(D)**. Different lowercase letters indicate significant differences between treatments (ANOVA, Tukey's test, $P < 0.05$). The data underlying this Figure can be found at https://doi.org/10.6084/m9.figshare.21692156.v1.

mass attacks, the host-derived oxygenated monoterpene *cis*-verbenol acts together with 2-methyl-3-buten-2-ol as an aggregation pheromone for *I. typographus* to promote mass attack on individual trees [28,54]. Beetles also utilize oxygenated monoterpenes to restrict the density of attack. Microbes lining the gallery walls or living in the beetle gut oxidize *cis*- and *trans*-verbenol into verbenone, which inhibits the attraction of both sexes to densely colonized trees [55–57]. In addition, mated male *I. typographus* produce ipsenol and ipsdienol, of which ipsenol acts as an anti-attractant [57]. Furthermore, an oxygenated monoterpene from host trees, 1,8-cineole, which is produced in higher amounts in resistant or MeJA-primed trees, inhibits

attraction of beetles to their pheromones [42,58,59]. Oxygenated monoterpenes are also used as reliable cues by parasitoids of bark beetles to locate their prey [35,60].

Here, we discovered that a blend of oxygenated monoterpenes emitted by *I. typographus*-associated fungi growing on agar medium amended with spruce bark or specific spruce monoterpenes is highly correlated with beetle attraction (Fig 6A and 6C). These same fungi were shown in a previous study [38] to produce mixtures of non-terpene aliphatic and aromatic volatiles when grown without sources of monoterpenes, and a blend of these aliphatic and aromatic compounds attracted newly emerged (callow) adult *I. typographus* [38]. These non-terpene compounds were also detected in the present study as major components of the volatile blend at later phases of fungal growth (S3–S7 Tables). Although we focused principally on the symbiont *G. penicillata* in our work, volatiles from other fungal symbionts were also investigated. *L. europhioides* and *E. polonica* emitted volatiles on SBA that were also attractive to adult *I. typographus* (S1 Fig), but the volatiles of *O. bicolor* and saprophytes such as *O. piceae* and *Trichoderma* sp. were not attractive.

The production of oxygenated monoterpenes by other microbes coinhabiting *I. typographus* galleries, such as fungal saprophytes, yeasts, and bacteria, has been reported before, but these microorganisms produce different metabolites of host tree monoterpenes (or the same metabolites in different ratios) [61–63] than those produced by symbiotic fungi as reported here. In addition, the blend of oxygenated monoterpenes emitted by *I. typographus* alone was distinct from that emitted by the fungal symbionts [64] (S12 Fig). Since oxygenated monoterpenes are major components of the volatile profiles of *I. typographus* fungal symbionts that are attractive to beetles (S2–S7 Tables and S9 Fig), it can be suggested that the oxygenated monoterpenes themselves are important in *I. typographus* attraction. Bioassays of individual oxygenated monoterpenes showed that two were attractive to beetles at specific concentrations, and two had no effect (Fig 6B and 6D). *I. typographus* was recently reported to be attracted to the fungi associated with the North American spruce bark beetle *D. rufipennis* when attacking a different species of spruce than *I. typographus* normally does in its home range. This attraction could also be due to oxygenated monoterpene profiles, although it is unclear if an association with these fungi would be beneficial to *I. typographus* [65].

## Oxygenated monoterpenes signal the presence of fungi to *I. typographus* and may modulate beetle colonization

The first chemical signals reported to mediate bark beetle colonization of their hosts were aggregation pheromones. Yet even in the presence of these pheromones, a large proportion of aggregating beetles that land on trees leave without tunneling into the bark [45,66,67]. This suggests that other cues may be needed to induce beetles to stay and bore into the bark. Indeed, bark beetles have been shown to respond to visual and chemical cues from host and nonhost species when selecting trees for colonization [28,38,44,68–70]. Based on our results, fungus-produced oxygenated monoterpenes might also belong to the list of colonization cues. For bark beetles, fungal metabolites can serve as indicators of host tree sectors where their fungal symbionts are already established. These compounds could also provide evidence for the ongoing metabolism of host tree defenses or signal that tree defenses have been overwhelmed, which could improve the success of bark beetle colonization.

Fungal volatiles also enhance the attraction of *I. typographus* to aggregation pheromones. Female *I. typographus* is known to use the aggregation pheromone mixture (2-methyl-3-buten-2-ol and *cis*-verbenol) to locate trees suitable for mating and oviposition [45,71]. Here, we showed that female *I. typographus* were more attracted at short range to a combination of the pheromone mixture plus fungal volatiles than to the pheromone mixture alone.

Similarly, a recent study reported that the presence of fungal volatiles increased the attraction of dispersing beetles to their pheromones in the field compared to traps with pheromones only [72]. Oxygenated monoterpenes and other fungal volatiles could provide information about the presence of fungal symbionts, which promote the successful development of their off-spring. Similarly, the pheromone component *cis*-verbenol, itself an oxygenated monoterpene produced by *I. typographus* from the host tree precursor α-pinene [73], provides information about the presence of other beetles, especially males. The lack of response of males to phero-mones in our experiments is not unexpected, as male *I. typographus* have been reported to be less responsive than females to high doses of pheromones in walking bioassays [71,74]. This behavior may help them avoid dense colonies of male conspecifics within a tree to reduce com-petition for mates and food.

Depending on how fast symbiotic fungi colonize the tree bark, fungal cues could benefit both epidemic and endemic beetle populations. During the endemic phase, beetles generally colonize stressed trees in a gradual fashion over months [66], and in this case, volatiles emitted from fungal symbionts introduced into the tree by early colonizers, in combination with beetle pheromones, could increase the probability for conspecifics to locate attacked trees, present at low densities in spruce stands. In contrast, during epidemics, vigorous trees are typically colo-nized by large numbers of bark beetles within a few days, which overwhelms their defenses. Since ophiostomatoid fungi have been reported to emit volatiles within 1 d following the infec-tion of detached bark plugs in the laboratory [75,76], these chemicals might function in attrac-tion during epidemics as well. However, during rapid natural attacks, it is unclear if fungi can colonize a tree sufficiently to emit behaviorally active volatiles. More research in this area is needed.

The oxygenated metabolites of host monoterpenes produced by fungal symbionts may not only attract bark beetles, but also stimulate them to tunnel in a fungus-colonized medium. *I. typographus* tunneled more in a diet with *G. penicillata* and β-pinene than with either fungus or monoterpene alone, perhaps because the fungus metabolized β-pinene to behaviorally active metabolites. However, this effect was not seen for α-pinene or bornyl acetate. Interestingly, both sexes showed similar tunneling preferences to fungi and monoterpenes in contrast to their different response to fungi and pheromones. The lack of sex-specific differences in tunneling could be due to the fact that the nutritional advantage of feeding on fungus-colo-nized spruce bark is beneficial to both male and female adults [17,18]. No other bioassays per-formed in this study showed sex-specific differences.

The proportion of oxygenated monoterpenes to total monoterpenes in the volatile blend of *G. penicillata* increased over the time course studied to nearly 50% at 12 d and nearly 70% at 18 d post-inoculation, a trend also observed for *L. europhioides* and *O. bicolor*. Thus, higher proportions of oxygenated monoterpenes may indicate older fungal infection sites and, hence, older beetle invasion sites that may be less attractive to newly arriving beetles due to crowding. The lack of attraction and even repellency of certain individual oxygenated monoterpenes, as seen in laboratory bioassays in our own and in previous studies [77] is consistent with this interpretation. Among the oxygenated monoterpenes already reported to inhibit *I. typogra-phus* attraction, verbenone, produced by microbial or autooxidation of the pheromone *cis*-ver-benol (or by direct oxidation of α-pinene), reduces aggregation of *I. typographus* in later phases of the attack on the tree [57,63,78].

Oxygenated monoterpenes are signals not only for bark beetles, but also for their natural enemies. Both beetle predators and parasitoids employ these compounds and other volatiles to locate bark beetle larvae hidden under the bark [79,80]. Specifically, a three-component blend comprising camphor, isopinocamphone, and terpinen-4-ol, all fungal metabolites of host tree monoterpene hydrocarbons, was reported to attract a coleopteran predator and several

hymenopteran parasitoids of *I. typographus* in the presence of host tree background signals [60,81,82]. Furthermore, the bark beetle predator, *Thanasimus formicarius*, possesses OSNs to detect oxygenated monoterpenes such as camphor and pinocamphone as well as several additional OSN classes detecting various oxygenated bark beetle pheromone compounds [83]. Thus, any benefit to the beetle arising from oxygenated monoterpene production by its symbiotic fungi may come at the cost of revealing its presence to natural enemies that employ these same volatiles to locate bark beetles.

## Formation of oxygenated derivatives may reduce monoterpene toxicity for *I. typographus*

The conversion of host tree monoterpene defenses by symbiotic fungi to oxygenated products may alleviate toxicity to bark beetles. Terpene-rich resins are a general defense of *P. abies* and other conifers against herbivores and pathogens [84–86]. Thus, it is not surprising that monoterpenes have exhibited toxicity to bark beetles in many studies [87–89]. Monoterpene hydrocarbons, such as α-pinene, are typically more toxic to beetles than host tree-produced oxygenated monoterpenes, such as bornyl acetate [90]. Hence, the oxidative transformations carried out by fungal symbionts described in this study could reduce toxicity to *I. typographus* through conversion to less poisonous derivatives. Such detoxification of host tree defenses could represent a significant benefit of fungal symbionts [91].

By contrast, oxygenated monoterpenes may be more toxic to fungi compared to monoterpene hydrocarbons [48,49]. Thus, the initial oxidation of monoterpene hydrocarbons may not constitute a detoxification unless it is a step towards further metabolism. The potential toxicity of oxygenated monoterpenes may explain why these substances are degraded further after initial oxidation by fungi specialized on conifers such as *Heterobasidion parviporum* and *Seridium cardinale* [48,49]. Another class of terpenes present in *P. abies* bark and other conifers host trees of bark beetles are diterpene resin acids, which were reported to be toxic to associated fungi in a North American bark beetle system [92].

## Other sources of oxygenated monoterpenes in spruce bark beetle interactions

Oxygenated monoterpenes emitted from trees attacked by *I. typographu*s may arise from sources other than fungal symbionts. The host tree *P. abies* itself synthesizes large amounts of bornyl acetate [93] and small amounts of 1,8-cineole [94]. In these compounds, the oxygen functions are incorporated during biosynthesis from basic precursors, whereas the products from fungal symbionts are presumably formed by oxidative modification of a previously formed monoterpene hydrocarbon skeleton. The compound *trans*-4-thujanol belongs to the latter group. We identified it as a *G. penicillata* metabolite of α- and β-pinene, but *trans*-4-thujanol may also be synthesized by the tree, although at low levels in *P. abies* bark [77]. As an alternative, this and other oxygenated monoterpenes could be produced via autoxidation of monoterpene hydrocarbons [95]. The oxidation of monoterpenes upon exposure to air could explain the release of low but readily detectible amounts of these compounds from uninfected control bark plugs in our and other studies. In the field, oxygenated monoterpenes other than bornyl acetate and 1,8-cineole have been detected from damaged *P. abies* trees when monoterpenes were exposed to air [96,97]. However, in the present study, the emission rate from uninfected bark plugs was much lower than from fungus-infected plugs, suggesting that microbial metabolism is a much more significant source of oxygenated monoterpenes than autoxidation (S9 Fig) [95]. Nevertheless, the tree itself cannot be ruled out as a source of any of the detected oxygenated monoterpenes, since *P. abies* cell suspension cultures have been reported to

oxidize added monoterpenes [94,95]. If *P. abies* produced monoterpenes as a defense response only when infected by fungi, it might be difficult to determine which organism was producing these compounds—the tree or the fungus—without identifying specific biosynthetic genes in one of the two genomes. More research is needed to assess the contribution of host trees to the oxygenated monoterpenes emitted in response to beetle tunneling or fungal infection.

Among microbial sources of oxygenated monoterpenes are several yeast species including *Hansenula holstii*, *H. capsulata*, and *Candida nitratophila*, which were isolated from *I. typographus* and produce terpinen-4-ol, α-terpineol, borneol, and *trans*-pinocarveol when grown in phloem medium or in α-pinene-supplemented medium [63]. In addition, several bacterial symbionts of bark beetles are capable of metabolizing monoterpenes into oxygenated derivatives, such as verbenols and verbenone [62]. Some bacteria coinhabit the beetle gallery together with ophiostomatoid fungi, but it is not known if these bacteria induce the production of oxygenated monoterpenes, either by producing them themselves or by interacting with fungi to form these compounds [98]. Another bark beetle species, *Polygraphus poligraphus*, which is sometimes found together with *I. typographus*, was shown to emit large amounts of terpinen-4-ol [99,100]. Intermediate amounts of α-terpineol, *cis*- and *trans*-4-thujanol were also identified from the hindgut as well as the entrance holes of this beetle's gallery and could be formed by microorganisms associated with this beetle species from host tree monoterpenes or by the beetle itself. Our experiments with fungus-free *I. typographus*, however, demonstrated that live beetles themselves did not produce high concentrations of any of the fungal biotransformation products identified in this study. Instead, beetles formed a range of other oxidation products (S12 Fig) that have been identified in previous studies [64,101]. In summary, our study brings together several lines of evidence showing that the oxygenated derivatives of spruce resin monoterpenes, which appear during *I. typographus* attack, are produced mainly by the metabolism of symbiotic fungi. The fungal origin of these compounds had been suggested previously [32] but not supported by experimental evidence until now.

## High selectivity of *I. typographus* olfactory neurons to oxygenated monoterpenes suggest their role in detecting symbiotic fungi

*I. typographus* possesses several classes of OSNs that were shown to detect the oxygenated monoterpenes produced by their fungal symbionts with notable specificity. For example, the isopinocamphone OSN class showed high specificity towards several monoterpene ketones produced by fungal symbionts, including (+)- and (−)-isopinocamphone, (+)- and (−)-pinocamphone, camphor, and pinocarvone, but not to monoterpene alcohols such as borneol and *trans*-pinocarveol. The absence of any response to monoterpene hydrocarbons indicates that this OSN is not tuned to detect the host tree itself, but rather organisms metabolizing the major host monoterpenes. The isopinocamphone OSN responded similarly as one recently reported OSN class from *I. typographus* that responded best to pinocarvone and camphor (OSN class named Pcn; isopinocamphone and pinocamphone were not tested) [97]. Our work shows that (+)-isopinocamphone is the primary ligand of this OSN class, based on its greater activity than the other active compounds. In addition, the response profile of this OSN class matches very well with that of the *I. typographus* odorant receptor 29 (ItypOR29), which recently was characterized in the oocytes of the African clawed frog, *Xenopus laevis* [102].

Likewise, we showed that a previously described verbenone-sensitive OSN class [41] also responds to *cis*- and *trans*-verbenol as well as α- and β-isophorone, compounds that are believed to arise from bark beetle metabolism of host tree terpenes [64]. Verbenone is produced from the verbenols by microbes that colonize gallery walls and beetle guts. Therefore, this OSN class appears to be tuned to signals from various ecological sources providing

information on bark beetle density as well as microbial establishment [45,56,96]. Since, as discussed above, individual oxygenated monoterpenes arise from different organisms, including fungi, other microbes, the host tree, and the beetles themselves, the presence of OSNs responding to different oxygenated monoterpenes may help the beetle identify the various life forms it encounters.

Another OSN class responded most sensitively to the monoterpene alcohol *trans*-4-thujanol, a fungal symbiont metabolite of α- and β-pinene. This OSN also responded to the fungal metabolites terpinen-4-ol and α-terpineol, as well as $C_8$ alcohols, but only at the highest doses tested, which extends prior results for this OSN [97] to other compounds from our greatly expanded test odor panel. Strong electroantennographic activity in *I. typographus* in response to these oxygenated monoterpenes has also been reported [96,97], and the response spectrum of this OSN class matches well with that of the receptor ItypOR23, which is evolutionarily related to the receptor ItypOR29 that detects isopinocamphone [102]. The specific responses of *I. typographus* bark beetles to oxygenated monoterpenes may arise not only from the selectivity of the OSN classes, but also due to further processing of the odor signal in the antennal lobe and other higher centers of the brain responsible for sensory integration (mushroom bodies and lateral horns) [103]. Moreover, since some of the oxygenated monoterpenes elicited secondary responses from other OSN classes, bark beetles like other insects may process olfactory information through combinatorial codes [70], which could lead to very specific responses to different compounds or mixtures.

## Conclusions

We have shown that fungal ectosymbionts vectored by *I. typographus* increased the emission of oxygenated monoterpenes from spruce bark due to fungal metabolism of host tree monoterpenes. These oxygenated volatiles were detected by several classes of *I. typographus* OSNs; some volatiles attracted beetles at specific concentrations, while others had no effect. Oxygenated monoterpenes appear to signal the presence of established fungi, but their effects on beetles are context-dependent and could vary with the physiological status of the fungi, the age, sex, and density of beetles, and the stage of attack [38,76]. Moreover, the ecological roles we have proposed for these oxygenated monoterpenes are based on laboratory assays with walking beetles, so studies under natural conditions are necessary to confirm our findings. These compounds may also be useful in integrated pest management strategies as attractants or repellents of bark beetles perhaps in combination with pheromones [104–106]. In this way, oxygenated monoterpenes and other microbial volatiles represent a rich source of untapped insect semiochemicals that can be exploited for protecting forests from devastating pest species such as *I. typographus*.

## 1. Materials and methods

### 1.1 Fungal strains and growth medium

The fungal strains used in this study have been previously described [38] (listed in S1 Table). In order to obtain spores from fungi, freshly inoculated PDA plates were incubated at room temperature for 15 to 20 d until the mycelium was old and dark. After 20 d, plates were kept briefly at 4°C to induce sporulation. Four to six 1 cm diameter mycelium plugs were removed from each plate and inoculated into 20 mL potato dextrose broth and incubated at 25°C at 150 rpm for 4 days. Once the broth was turbid, the spores were filtered using a 40-μm cell strainer (Greiner Bio-One, Frickenhausen, Germany), and the filtrate was spun down at 4,200 rcf for 10 min to precipitate the spores. The supernatant was discarded, and the spore suspension was

washed three times with autoclaved water and then stored at 4°C until used. The spore suspension prepared using this method was viable for several months when stored at 4°C.

## 1.2 Bark beetle rearing

Bark beetles were reared and stored in the laboratory as described [38]. The starting beetle culture was obtained from an infested tree in October 2017 near Jena, Thuringia, Germany. Beetles were reared throughout the year in the laboratory in freshly cut spruce logs (ca. 30 cm diameter × 50 cm height) placed in an environmental chamber set at 25°C throughout the day, 65% relative humidity, and a photoperiod of 20 h per day. Beetles emerged from breeding logs after ca. 35 d and were collected manually. Emerged adults were sexed based on the bristle density on their pronotum [107] and stored separately in Falcon tubes lined with moist paper at 4°C at least for a week before using them in bioassays. Adult beetles were used only once in bioassays.

## 1.3 Spruce bark diet

The outer bark of a freshly cut mature tree was scraped off gently using a drawing knife, and the inner bark (phloem) was carefully peeled off using a chisel and immediately stored at −80°C. The bark was cut into small pieces and ground to a fine powder in vibratory micro mill (Pulverisette 0, Fritsch GmbH, Idar-Oberstein, Germany). The instrument was precooled with liquid nitrogen, and bark pieces were pulverized at an amplitude of 2.0 for ca. 10 min with addition of liquid nitrogen every 2 min to prevent thawing. The ground powder was stored in Falcon tubes at −80°C until used for diet preparation. For preparing spruce bark diet, 7% powdered inner spruce phloem (w/v) was added to 4% Bactoagar (Roth) and heat sterilized at 121°C for 20 min.

## 1.4 Identification and quantification of headspace volatiles of fungal symbionts

Collection and analysis of headspace volatiles for *G. penicillata* and *Trichoderma* sp. comparison (S9 Fig) is described in S2 Method. Norway spruce bark plugs of approximately 28 mm diameter were removed from a freshly felled tree in July 2017 and single bark plugs placed inside sterile 250 mL glass bottles for volatile collection. Before removing the bark plugs, the surface of the bark and the cork borer were sterilized by thorough spraying with 70% ethanol in a laminar hood. A 100-μL quantity of spore suspension ($1 * 10^6$ cells mL$^{-1}$), prepared as described above, was added to the exposed section of the bark, and autoclaved water was added to the control treatment. Each treatment was replicated four times including the control. The glass bottle was secured tightly and incubated at 25°C for 4 d. After 4 d, activated charcoal-filtered air was passed into the bottle inlet at the rate of 50 mL min$^{-1}$, and the outlet air was funneled through a SuperQ adsorbent filter (150 mg) for 4 h. Afterwards, the filters were eluted with 200 μL dichloromethane spiked with 10 ng μL$^{-1}$ nonyl acetate (Sigma Aldrich) as an internal standard and stored at −20°C. The spruce bark plugs were oven dried at 80°C for 6 h after the experiment, and the dry weight was measured.

The eluted volatile samples were subjected to GC–MS analysis for identification and GC-FID analysis for quantification using an Agilent 6890 series GC (Agilent, Santa Clara, CA, USA) (injection, 1 μl splitless; flow, 2 ml min$^{-1}$; temperature, 45 to 180°C at 6°C min$^{-1}$ and then to 300°C at 100°C min$^{-1}$ for 10 min) coupled either to an Agilent 5973 quadrupole mass selective detector (interface temperature 270°C, quadrupole temperature 150°C, source temperature 230°C; electron energy 70 eV) or a flame ionization detector (FID, temp. 300°C). The constituents were separated on a DB-5MS column (Agilent (30 m × 0.25 mm × 0.25 μm)),

with He (MS) or $H_2$ (FID) as carrier gas. The GC–MS and GC-FID analyses used identical columns and temperature programs so that the chromatograms could be overlaid to facilitate identification and quantification. Peaks arising from the solvent and collection containers were identified by blank runs and excluded from the analysis. The identity of each peak was determined by comparing its mass spectra and retention times to those of reference libraries (NIST98 and Wiley275) and authentic standards. The amount of each compound was calculated from the peak area obtained from the FID detector relative to the internal standard and standardized to the spruce bark dry weight.

## 1.5 Time series headspace volatile collection

For time series volatile analysis, spruce bark plugs (10 mm diameter) were removed using a cork borer from a freshly felled spruce tree in October 2016. Each spruce bark plug was placed in a 15-mL clear glass vial (Supelco-Sigma-Aldrich), and 50-μL spore suspension ($1 * 10^6$ cells $mL^{-1}$), prepared as described above, was added to treatment plugs while control plugs received sterile water. The headspace volatiles were captured on three polydimethylsiloxane (PDMS) sorbent silicone tubes (0.5 cm), which were hung in each glass vial using a manually crafted metal hook attached to the bottom of PTFE/silicone septa in the screw cap [108]. The headspace volatiles were collected from each treatment for 2 h at 4, 8, 12, and 18 d after inoculation. After sampling, silicone tubes were placed in 1.5 mL brown glass vials and stored at −20˚C until analysis.

Volatiles collected on PDMS tubes were analyzed using a GC-2010 plus gas chromatograph coupled to a MS-QP2010 quadrupole mass spectrometer equipped with a TD-20 thermal desorption unit (Shimadzu, Japan) and a GC Cryo-Trap filled with Tenax. A single tube was placed in a 89-mm glass thermal desorption tube and desorbed at a flow rate of 60 mL $min^{-1}$ for 8 min at 200˚C under a stream of $N_2$ gas. The desorbed substances were focused on a cryogenic trap at −20˚C. The Tenax adsorbent was heated to 230˚C, and the analytes were injected using split mode (1:100) onto a Rtx-5MS GC column (30 m × 0.25 mm × 0.25 μm) with helium as carrier gas. Compounds were identified as above (1.5) from authentic standards and libraries, and the area of each peak obtained using GC–MS post run analysis software from Shimadzu. Peaks arising from contaminants from the solvent, medium, or bark plug container were identified by blank runs and excluded from the analysis. The PLS-DA plot in Fig 2A was generated by using MetaboAnalyst 3.0 software with normalized GC–MS data (both log transformed and range scaled) [109].

## 1.6 Biotransformation of host tree compounds

Experiments were conducted in 9 cm Petri dishes containing 2% PDA supplemented with test solutions. The tested compounds were (−)-α-pinene, (+)-α-pinene, (−)-β-pinene, myrcene, γ-terpinene, terpinolene, sabinene, camphene, $p$-cymene, and (−)-bornyl acetate. Sources of these compounds are given in S1 Table. Compounds were added after dissolving in dimethyl sulfoxide (DMSO). These were then added into PDA (maintained at 55 to 57˚C using a water bath) to reach a concentration of 0.5 mg $mL^{-1}$ before pouring into Petri dishes. A 5-mm agar plug containing a fungal colony was placed in the center of each dish and sealed tightly using several layers of Parafilm before incubation at 25˚C in darkness for 6 d. Each treatment was replicated four times, and for the control, the PDA contained only DMSO plus monoterpene. The headspace volatiles were collected after 4 d using three PDMS tubes, which were mounted on sterile metal wires and imbedded in PDA for 1 h and stored at −20˚C. The identification and quantification of compounds were conducted in the same way as reported for the time series (section 1.5). The headspace volatiles from fungus grown on PDA enriched with

myrcene, γ-terpinene, terpinolene, camphene, and *p*-cymene did not yield detectable amounts of monoterpene transformation products on analysis.

To identify if symbiotic fungi can reduce the amount of monoterpenes in their substrate, fungi were grown on PDA enriched with 0.5 mg g$^{-1}$ (−)-α-pinene, (+)-α-pinene, (−)-β-pinene, and (−)-bornyl acetate as described above. Control plates contained only DMSO and the tested monoterpene. After 4 d, three plugs of 6 mm diameter were removed, weighed, and transferred to 1.5 ml sterile glass vials. Agar plugs were homogenized using sterile plastic pestles and 1 ml hexane (extraction solvent) spiked with 10 ng μL$^{-1}$ nonyl acetate was added and samples were vortexed for 30 s. Supernatants were transferred to new vials and stored at −20˚C until identification and quantification by GC–MS and GC-FID (1.4). Data analysis was identical to that reported above.

## 1.7 Chemical synthesis of (+)-isopinocamphone and β-isophorone

**(+)-isopinocamphone** ((1*R*,2*R*,5*S*)-2,6,6-Trimethylbicyclo[3.1.1]heptan-3-one) was synthesized as described in [110,111]. A mixture of (1*R*,2*R*,3*R*,5*S*)-(−)-isopinocampheol (200 mg, 1.30 mmol, Sigma-Aldrich) and Dess-Martin-periodinane (825 mg, 1.95 mmol) in anhydrous CH$_2$Cl$_2$ (15 mL) was stirred at room temperature for 1 h, followed by the addition of water and sat. aq. NaHCO$_3$ solution. The mixture was extracted twice with methyl *t*-butyl ether. The organic phase was washed with brine, dried over anhydrous Na$_2$SO$_4$, and concentrated in vacuo. The residue was purified by short-path chromatography using an SPE cartridge (Chromabond SiOH, 6 mL, 500 mg, Macherey-Nagel, *n*-hexane:EtOAc = 10:1) to yield (+)-isopinocamphone (159 mg, 1.04 mmol, 80%). NMR measurements were carried out on a Bruker Avance AV-500HD spectrometer, equipped with a TCI cryoprobe using standard pulse sequences as implemented in Bruker Topspin ver. 3.6.1. (Bruker Biospin GmbH, Rheinstetten, Germany). Chemical shifts were referenced to the residual solvent signals of acetone-$d_6$ ($\delta_H$ 2.05/ $\delta_C$ 29.84) or CDCl$_3$ ($\delta_H$ 7.26/ $\delta_C$ 77.16), respectively.

$^1$H-NMR (500 MHz, CDCl$_3$) $\delta$ ppm: 2.64 (*ddd*, *J* = 18.6/3.0/3.0 Hz, 1H), 2.62 (*dddd*, *J* = 10.1/6.2/6.1/3.0 Hz, 1H), 2.52 (*bd*, *J* = 18.6 Hz, 1H), 2.46 (*dq*, *J* = 7.3/1.9 Hz, 1H), 2.12 (*ddd*, *J* = 9.1/6.1/3.0 Hz, 1H), 2.06 (*ddd*, *J* = 6.2/6.2/1.9 Hz, 1H), 1.31 (*s*, 3H), 1.21 (*d*, *J* = 7.3 Hz, 3H), 1.19 (*bd*, *J* = 10.1 Hz, 1H), 0.88 (*s*, 3H). $^{13}$C-NMR (125 MHz, CDCl$_3$) $\delta$ ppm: 215.9, 51.7, 45.3, 45.1, 39.4, 39.1, 34.8, 27.4, 22.2, 17.3. GC–MS t$_R$: 13.6 min. EI-MS (70 eV): *m/z* (%) 152 (14), 110 (15), 95 (44), 83 (89), 69 (97), 55 (100), 41(64) (S13 Fig).

**β-isophorone** (3,5,5-Trimethyl-3-cyclohexen-1-one). β-Isophorone was synthesized from α-isophorone (Acros Organics, Fair Lawn, NJ, USA) following published methods [112]. Since β-isophorone was very unstable during column chromatography [113], the compound was used for the bioassay without purification. The purity of β-Isophorone was 91% with 6% of α-isophorone as assessed by NMR analysis. $^1$H-NMR (500 MHz, acetone-$d_6$) $\delta$ ppm: 5.44 (*s*, 1H), 2.69 (*bs*, 2H), 2.27 (*s*, 2H), 1.70 (*bs*, 3H), 1.00 (*s*, 6H). $^{13}$C-NMR (125 MHz, acetone-$d_6$) $\delta$ ppm: 209.3, 133.0, 130.6, 53.6, 44.2, 36.9, 29.8, 22.8. GC–MS t$_R$: 10.3 min. EI-MS (70 eV): *m/z* (%) 138 (69), 123 (68), 96 (99), 95 (84), 81 (100) (S14 Fig).

Purity of the synthesized compounds was also determined using the following GC–MS program: injection, 1 μl splitless; flow, 2 ml min$^{-1}$; temperature, 45˚C (held for 2 min) to 250˚C with 6˚C min$^{-1}$.

## 1.8 Electrophysiology

Laboratory reared adult beetles from the same German culture that were used in bioassays were used for electrophysiological SSRs using tungsten microelectrodes according to established methodology [38,41] with the equipment previously described from Syntech,

Kirchzarten, Germany [114]. The odor panel comprising 92 compounds consisted of beetle pheromones, host tree, nonhost tree, and fungal compounds ([38]; S2 Table). Both major and minor fungal volatiles identified during the chemical analysis were included in the odor panel. All odors were dissolved in odorless paraffin oil (w/v). SSR traces were analyzed as described [105] using Autospike 3.0 (Syntech). Males and females were initially screened for responses to the odor panel using a high-stimulus dose (10 μg on filter paper placed inside capped standard Pasteur pipette odor cartridges; [114]). OSN classes shown to primarily respond to fungus-derived oxygenated monoterpenes were subsequently studied in dose–response experiments with active stimuli diluted in 10-fold steps and tested from lowest to highest dose with the least active ligands tested first at each dose. To reduce variation due to odor depletion, stimulus cartridges were used for a maximum of 8 stimulations during screening and 2 stimulations during dose–response tests [115].

### 1.9 Trap bioassay

The arena used was described in [38]. In this apparatus, adult beetles had to make their choice through olfaction and not by contact cues. Fungi were inoculated on SBA-based diet and incubated at 25°C for 4 d. With the help of a cork borer (10 mm diameter), bark plugs with or without fungus were inserted into circular plastic cups (1.8 cm height * 1.8 cm diameter) facing each other. Two beetles were placed at the center of each arena, and the olfactometer was placed inside a laminar flow cabinet in darkness. Each experiment was replicated at least 25 times on the same day with 2 beetles per replicate. Treatments were randomly assigned to cups, and cups were rinsed thoroughly using acetone between experiments. Adult beetles were used only once. The choice of beetles was determined after 6 h by counting the number of beetles inside the cups and represented as percentage choice (percentage of insects responding to either control traps or treatment traps or no response). When two SBA control plugs were tested simultaneously, adult beetles showed no preference for traps on one side of the arena versus the other side. Preliminary experiments showed that the sex of the beetle did not influence the olfactory response towards fungus grown either alone or in the diet enriched with monoterpenes. Therefore, two beetles were randomly chosen for trap bioassays.

For bioassays using terpenes, stock solutions were prepared by dissolving the compounds in DMSO, which were then added to 7% SBA to a final concentration of 0.05 to 1 mg g$^{-1}$. To determine the response of adult beetles to (−)-β-pinene and (−)-bornyl acetate amended diet containing the fungus, *G. penicillata* was used as this species emitted higher amounts of biotransformation products compared to other fungi. Controls were treated with DMSO plus monoterpene (no fungus) or DMSO plus *G. penicillata* (no monoterpene). Approximately 7% SBA plugs (10 mm) supplemented with monoterpenes or plugs containing *G. penicillata* were placed in the control cups, and *G. penicillata*-colonized plugs from monoterpene-enriched medium were placed in the treatment cups. The volatile emission from each control and treatment plug used in the bioassays was determined using PDMS tubes as adsorbents and analyzed as described previously (section 1.5). For bioassays with synthetic compounds, stock solutions of authentic standards were prepared by dissolving them in mineral oil (w/v) and further diluted in log$_{10}$ steps by dissolving in mineral oil. Approximately 10 μL was applied to 10 mm Whatman filter paper laid on the top of SBA plugs placed inside the cups. Control traps were treated with 10 μL paraffin oil. For the experiment with pheromone blend in the presence of *G. penicillata* volatiles, *G. penicillata*-colonized spruce bark plugs were placed in treatment cups, and 10 μL of a pheromone mixture (*cis*-verbenol:2-methyl-3-buten-2-ol in the ratio of 1:50 diluted 1:100 in paraffin oil) was applied to filter paper as described above. Control cups were treated with 10 μL of the pheromone mixture. Males and females were tested separately

in this experiment as *I. typographus* bark beetles show sex-specific responses to their pheromone [68,114].

## 1.10 Tunneling behavior bioassay

*I. typographus* tunneling behavior was studied using a protocol modified from [92]. We tested beetles in $35 \times 10$ mm Petri dishes (Greiner Bio-one, Frickenhausen, Germany) filled with ca. 3 ml of spruce bark diet. The spruce bark diet was prepared as before with some modifications: 7% (w/v) spruce inner bark powder was mixed with 1% fibrous cellulose (Sigma), 2% glucose (Roth), and 4% Bactoagar (Roth) in water and autoclaved for 20 min at 121°C. Before pouring the medium into the Petri dishes, the medium was mixed with 2% solvent (DMSO: ethanol, 1:1) with 1 mg g$^{-1}$ of various monoterpenes ((−)-α-pinene, (−)-β-pinene, and (−)-bornyl acetate) and solvent only as a control. For treatment with fungus, 5 µl spore suspension of *G. penicillata* ($1 \times 10^6$ cells mL$^{-1}$) was added to the center of Petri dishes containing monoterpene-enriched media or solvent controls and incubated at 25°C for 4 d. A single beetle was introduced per plate, and the plates were sealed with Parafilm and kept in the environmental chamber for 48 h under conditions described above (section 1.2). The beetles were monitored for their tunneling activity after 2, 4, 6, 24, and 48 h with tunneling recorded as a binary event. If beetles were inside the media, it was noted as 1 and, if outside, noted as 0. After 48 h, tunnel lengths made by beetles in each plate were measured using Image J software. Each treatment was replicated with 15 male and female beetles.

## 1.11 Data analysis

IBM SPSS Statistics V25.0 was used to analyze the volatile differences between treatments (*E. polonica*-, *G. penicillata*-, *L. europhioides*-, *O. bicolor*-, and *O. piceae*-treated bark samples and untreated control). Data were log-transformed to meet the assumptions of normal distribution, as needed. Concentrations (in dry weight (ng h$^{-1}$ mg$^{-1}$)) of all individual compounds assigned to monoterpenes (MTs) or oxygenated MTs were combined and subjected to a *t* test or Welch's for estimating differences between control and *G. penicillata* (Fig 2). Additionally, separate ANOVAs for all individual compounds in each group were also performed (S2 Table). For volatile time course samples, a separate ANOVA test was performed for all individual compounds and compound groups from each fungus with time intervals as an independent factor (S3–S7 Tables). All ANOVA tests were followed by Tukey's post hoc tests to test for differences among treatment combinations. For behavioral bioassays, the CI values from each experimental group were analyzed by Wilcoxon's signed ranked test to compare the differences between control and treatment samples. Binary data from bark beetle tunneling assays were subjected to multiple logistic regression to analyze independent variables such as MT, sex, and fungus that influence the tunneling activity of beetles (dependent variable) in the medium. During data analysis, the male was coded as 1 and female as 0, the presence of fungus coded as 1 and absence of fungus as 0, tunneling inside the medium coded as 1 and not tunneling or staying outside the medium as 0. After testing all possible independent variables and their interactions among them, the following best-fitted logistic regression model was created to predict the odds of beetles tunneling in the different media.

Ln [odds] (tunneling odds) = β0 + β1 * compound + β2 * sex + β3 * fungus

Here, β0 is constant, whereas β1, β2, and β3 are logistic coefficients or estimates for the parameters for compound, sex, and fungus, respectively. The strength of association between beetle tunneling odds and effect of MTs or sex or fungus is expressed as odds ratios (OR = exp$^β$) where OR < 1 indicates a negative relationship between the two events, i.e., the tunneling event is less likely to happen in response to a selected independent variable (coded

as 1) in comparison with its base group (coded as 0), OR = 1 indicates no relationship between two events, OR > 1 shows positive relationship between two events.

## Supporting information

**S1 Fig. Adult beetles prefer spruce bark agar (SBA) inoculated with two species of symbiotic fungi over uninoculated SBA.** Adult beetles did not prefer *O. bicolor*, *O. piceae*, and *Trichoderma* sp., the latter two species are saprophytes. Deviation of response indices against zero was tested using Wilcoxon's test (*n* = 20 or 25). The data underlying this Figure can be found at https://doi.org/10.6084/m9.figshare.21692156.v1.
(TIF)

**S2 Fig. Volatile emission pattern differed between spruce bark inoculated with different fungi and uninfected bark 4 d after inoculation, as depicted in a sparse partial least squares discriminant analysis (sPLS-DA).** Analysis was performed using 59 compounds listed in S2 Table. Principal components (PC1 and PC2) explain 41.4% and 11.2% of the total variation, respectively, and ellipses denote 95% confident intervals around each species. The sPLS-DA plot was generated by using MetaboAnalyst 3.0 software with normalized data (both log transformed and range scaled). The data underlying this Figure can be found at https://doi.org/10.6084/m9.figshare.21692156.v1.
(TIF)

**S3 Fig. Changes in volatile emission profiles of fungal-infested vs. uninfested spruce bark over an 18-d time course for three other *I. typographus* symbiotic fungi besides *G. penicillata*. Compounds are classified into six groups according to chemical structures.** Complete volatile emission data by compound and time point for each fungal species are given in S3–S6 Tables. (*n* = 3 or 5). The data underlying this Figure can be found at https://doi.org/10.6084/m9.figshare.21692156.v1.
(TIF)

**S4 Fig. Volatile metabolites of (−)-β-pinene produced by two fungal symbionts (*L. europhioides* and *O. bicolor*) of *I. typographus* growing on potato dextrose agar.** Isopinocamphone was the major biotransformation product (*n* = 4 or 5). *E. polonica* produced no detectable products. ND, not detected. The data underlying this Figure can be found at https://doi.org/10.6084/m9.figshare.21692156.v1.
(TIF)

**S5 Fig. Metabolites of (−)-β-pinene produced by various *I. typographus* symbiotic fungi growing on potato dextrose agar after this monoterpene was administered to cultures of each species.** Amounts of metabolites were determined after hexane extraction of the agar. Error bars represent SEM (*n* = 5 or 11). ND, not detected. Different lowercase letters denote significant differences between treatments (ANOVA, Sidak's test; $P < 0.05$). Fungal abbreviations: *E. polonica* (Ep), *L. europhioides* (Le), *G. penicillata* (Gp), *O. bicolor* (Ob). The data underlying this Figure can be found at https://doi.org/10.6084/m9.figshare.21692156.v1.
(TIF)

**S6 Fig. Metabolites of (−)-α-pinene produced by various *I. typographus* symbiotic fungi growing on potato dextrose agar after this monoterpene was administered to cultures of each species.** Amounts of metabolites were determined after hexane extraction of the agar. Error bars represent SEM (*n* = 5 to 12). ND, not detected. Different lowercase letters denote significant differences between treatments (ANOVA, Sidak's test; $P < 0.05$). Fungal abbreviations: *E. polonica* (Ep), *L. europhioides* (Le), *G. penicillata* (Gp), *O. bicolor* (Ob). The data

underlying this Figure can be found at https://doi.org/10.6084/m9.figshare.21692156.v1.
(TIF)

**S7 Fig. Metabolites of (+)-α-pinene produced by various *I. typographus* symbiotic fungi growing on potato dextrose agar after this monoterpene was administered to cultures of each species.** Amounts of metabolites were determined after hexane extraction of the agar. Error bars represent SEM ($n$ = 5 or 13). ND, not detected. Different lowercase letters denote significant differences between treatments (ANOVA, Sidak's test; $P < 0.05$). Fungal abbreviations: *E. polonica* (Ep), *L. europhioides* (Le), *G. penicillata* (Gp), *O. bicolor* (Ob). The data underlying this Figure can be found at https://doi.org/10.6084/m9.figshare.21692156.v1.
(TIF)

**S8 Fig. Metabolites of (−)-bornyl acetate produced by various *I. typographus* symbiotic fungi growing on potato dextrose agar after this monoterpene was administered to cultures of each species.** Amounts of metabolites were determined after hexane extraction of the agar. Error bars represent SEM ($n$ = 5 or 13). ND, not detected. Different lowercase letters denote significant differences between treatments (ANOVA, Sidak's test; $P < 0.05$). Fungal abbreviations: *E. polonica* (Ep), *L. europhioides* (Le), *G. penicillata* (Gp), *O. bicolor* (Ob). The data underlying this Figure can be found at https://doi.org/10.6084/m9.figshare.21692156.v1.
(TIF)

**S9 Fig. Relative proportion of oxygenated monoterpenes produced by the bark beetle symbiont *G. penicillata*, a saprophyte *Trichoderma* sp. and a fungus-free control potato dextrose agar medium amended with a mix of spruce monoterpenes (see S8 Table).** The data underlying this Figure can be found at https://doi.org/10.6084/m9.figshare.21692156.v1.
(TIF)

**S10 Fig. Response spectra of olfactory sensory neuron (OSN) classes (originally characterized in [60]) with primary responses to (A) (+)-α-pinene ($n$ = 14), (B) *p*-cymene ($n$ = 9), and (C) Δ3-carene ($n$ = 4) to their respective most active ligands at the 10-μg screening dose (ligands eliciting average responses <20 Hz are not shown).** In addition to responses to the primary ligands, which are monoterpene hydrocarbons, these OSN classes show comparatively strong secondary responses to oxygenated monoterpenes produced by symbiotic fungi from host tree monoterpenes. Error bars represent SEM. The data underlying this Figure can be found at https://doi.org/10.6084/m9.figshare.21692156.v1.
(TIF)

**S11 Fig. Adult beetles did not discriminate between spruce bark agar (SBA) enriched with monoterpenes and unenriched SBA.** Error bars represent SEM ($n$ = 25 for each trial). The data underlying this Figure can be found at https://doi.org/10.6084/m9.figshare.21692156.v1.
(TIF)

**S12 Fig. Concentration of the oxygenated monoterpenes (a) *trans*-verbenol, (b) *cis*-verbenol, (c) borneol, (d) myrtanal, (e) myrtenol, (f) verbenone, (g) *trans*-myrtanol, (h) perillaldehyde, (i) nopinone, and (j) pinocarvone produced by live and dead male *I. typographus* fumigated with the mix of spruce monoterpenes listed in S8 Table.** The method for chemical analysis of beetles is in the supplementary methods (S3 Method). The data underlying this Figure can be found at https://doi.org/10.6084/m9.figshare.21692156.v1.
(TIF)

**S13 Fig. Confirmation of the structure of synthesized (+)-isopinocamphone by NMR. (A)** $^1$H and $^{13}$C signal assignments in deuterated chloroform (CDCl$_3$), **(B)** $^1$H NMR spectrum in

CDCl$_3$, **(C)** phase-sensitive heteronuclear single quantum coherence (HSQC) in CDCl$_3$, **(D)** heteronuclear multiple bond correlation (HMBC) in CDCl$_3$, **(E)** correlated spectroscopy (COSY) in CDCl$_3$, and **(F)** $^{13}$C attached proton test (APT) in CDCl$_3$.
(TIF)

**S14 Fig. Confirmation of the structure of synthesized β-isophorone by NMR. (A)** $^1$H and $^{13}$C signal assignments in acetone-$d_6$, **(B)** $^1$H NMR spectrum in acetone-$d_6$, purity- 91%, **(C)** phase-sensitive heteronuclear single quantum coherence (HSQC) in acetone-$d_6$, **(D)** heteronuclear multiple bond correlation (HMBC) in acetone-$d_6$, **(E)** correlated spectroscopy (COSY) in acetone-$d_6$, and **(F)** $^{13}$C spectrum in acetone-$d_6$.
(TIF)

**S15 Fig.** Scanning electron micrographs of (**A**) an elytron of an untreated bark beetle showing (**B**) spores of an ophiostomatoid fungus in the elytral pit, (**C**) an empty elytral pit of a fungus-free beetle, and (**D**) spore mass of *G. penicillata* in the elytral pit of a fungus-free beetle reinoculated with this fungal species.
(TIF)

**S1 Table. Purity, source, and biological origins of chemicals used in the experiments.**
(DOCX)

**S2 Table. Emission of volatile organic compounds identified from the headspace collection of fresh spruce bark 4 d after inoculation with different fungi.** Analyses were conducted using GC-FID. Compounds were identified by GC–MS analyses run in parallel. Compounds with significant *P* values are highlighted in **bold.** The data underlying this Table can be found at https://doi.org/10.6084/m9.figshare.21692156.v1.
(DOCX)

**S3 Table. Relative amounts (mean ± SE, *n* = 3) of volatiles from uninoculated bark detected after various time periods (4, 8, 12, and 18 d) from the beginning of an experiment with fungal inoculation.** Data from the control uninfected treatment are presented here. Data for fungal treatments are given in S3–S6 Tables. Volatiles were collected on polydimethylsiloxane tubes for 2 h and were subjected to GC–MS analysis (see Materials and methods section for details). ND, not detected, NA, not analyzed, TR, trace amounts (<500 TIC counts). The data underlying this Table can be found at https://doi.org/10.6084/m9.figshare.21692156.v1.
(DOCX)

**S4 Table. Relative amounts (mean ± SE, *N* = 4–5) of volatiles detected at various time periods after inoculation of fresh spruce bark with *E. polonica* (4, 8, and 12 d).** Volatiles were collected on polydimethylsiloxane tubes for 2 h and were subjected to GC–MS analysis (see Materials and methods section for details). ND, not detected, NA, not analyzed, TR, trace amounts (<500 TIC counts). The data underlying this Table can be found at https://doi.org/10.6084/m9.figshare.21692156.v1.
(DOCX)

**S5 Table. Relative amounts (mean ± SE, *N* = 5) of volatiles detected at various time periods after inoculation of fresh spruce bark with *G. penicillata* (4, 8, 12, and 18 d).** Volatiles were collected on polydimethylsiloxane tubes for 2 h and were subjected to GC–MS analysis (see Materials and methods section for details). ND, not detected, NA, not analyzed, TR, trace amounts (<500 TIC counts). The data underlying this Table can be found at https://doi.org/10.6084/m9.figshare.21692156.v1.
(DOCX)

**S6 Table. Relative amounts (mean ± SE, *N* = 5) of volatiles detected at various time periods after inoculation of fresh spruce bark with *L. europhioides* (4, 8, 12, and 18 d).** Volatiles were collected on polydimethylsiloxane tubes for 2 h and were subjected to GC–MS analysis (see Materials and methods section for details). ND, not detected, NA, not analyzed, TR, trace amounts (<500 TIC counts). The data underlying this Table can be found at https://doi.org/10.6084/m9.figshare.21692156.v1.
(DOCX)

**S7 Table. Relative amounts (mean ± SE, *N* = 5) of volatiles detected at various time periods after inoculation of fresh spruce bark with *O. bicolor* (4, 8, 12, and 18 d).** Volatiles were collected on polydimethylsiloxane tubes for 2 h and were subjected to GC–MS analysis (see Materials and methods section for details). ND, not detected, NA, not analyzed, TR, trace amounts (<500 TIC counts). The data underlying this Table can be found at https://doi.org/10.6084/m9.figshare.21692156.v1.
(DOCX)

**S8 Table. Composition of synthetic monoterpene mixture used in bioassays.** [$]The purity of each compound was calculated from GC–MS analysis.
(DOCX)

**S9 Table. Average colony forming units (CFUs/mL) from untreated, fungus-free, and fungus-free *G. penicillata*-reinoculated *I. typographus* bark beetles (*n* = 5 or 6 beetles).** Wash, supernatant from beetles immersed in 0.05% Triton X in 500 μL PBS buffer (pH 7.4); lysate, crushed beetles in 500 μL PBS buffer (pH 7.4); NP, not present. The data underlying this Table can be found at https://doi.org/10.6084/m9.figshare.21692156.v1.
(DOCX)

**S10 Table. Colony forming units (CFUs/mL) obtained from bark beetle gallery samples infested by fungus-free beetles, and fungus-free beetles reinoculated with *G. penicillata*, and untreated control beetles.** Approximately 300 mg of bark samples were dissolved in 1 mL PBS buffer solution, and dilutions were plated on PDA. NP, not present. [$]Only one gallery sample was tested due to low sample availability.
(DOCX)

**S1 Method. Preparation of bark beetle diet for eliminating fungal symbionts.**
(DOCX)

**S2 Method. Analysis of *G. penicillata* and *Trichoderma* sp. headspace volatiles.**
(DOCX)

**S3 Method. Beetle pheromone analysis.**
(DOCX)

## Acknowledgments

We thank Dr. Hong-Lei Wang for assistance in identification of terpene metabolites of bark beetles and Erling Jirle for his help to obtain materials for rearing bark beetles in Lund. We thank Bettina Raguschke and Daniel Veit for their assistance in the laboratory. We are grateful for the support of Stefan Engeter from ThüringenForst for permitting us to harvest spruce trees for the purpose of maintaining beetle culture in Jena. We also thank Dr. Jothi Kumar Yuvaraj for his assistance in single sensillum recordings, and Dr. Christian Schiebe, Prof. Rikard Unelius, and Prof. Fredrik Schlyter for providing some of the stimuli used in SSR. We

thank Ola Gustafsson, Microscopy Facility, Department of Biology, Lund University, for help with scanning electron microscopy.

## Author Contributions

**Conceptualization:** Dineshkumar Kandasamy, Martin N. Andersson, Almuth Hammerbacher, Jonathan Gershenzon.

**Data curation:** Dineshkumar Kandasamy.

**Formal analysis:** Dineshkumar Kandasamy, Rashaduz Zaman.

**Funding acquisition:** Martin N. Andersson, Jonathan Gershenzon.

**Investigation:** Dineshkumar Kandasamy, Rashaduz Zaman, Yoko Nakamura, Tao Zhao.

**Methodology:** Dineshkumar Kandasamy, Martin N. Andersson.

**Project administration:** Dineshkumar Kandasamy, Martin N. Andersson, Almuth Hammerbacher, Jonathan Gershenzon.

**Resources:** Henrik Hartmann, Martin N. Andersson, Jonathan Gershenzon.

**Supervision:** Dineshkumar Kandasamy, Martin N. Andersson, Almuth Hammerbacher, Jonathan Gershenzon.

**Validation:** Dineshkumar Kandasamy.

**Visualization:** Dineshkumar Kandasamy.

**Writing – original draft:** Dineshkumar Kandasamy.

**Writing – review & editing:** Henrik Hartmann, Martin N. Andersson, Almuth Hammerbacher, Jonathan Gershenzon.

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
