## [Editor Report · Decision Letter 0]

18 Oct 2022

Dear Jonathan, 

Thank you for submitting your revised manuscript entitled "Bark beetles locate fungal symbionts by detecting volatile fungal metabolites of host tree resin monoterpenes" for consideration as a Research Article by PLOS Biology.

Your revisions have now been evaluated by the PLOS Biology editorial staff, and I'm writing to let you know that we would like to send your submission out for re-review.

However, before we can send your manuscript back to reviewers, we need you to complete your submission by providing the metadata that is required for full assessment. To this end, please login to Editorial Manager where you will find the paper in the 'Submissions Needing Revisions' folder on your homepage. Please click 'Revise Submission' from the Action Links and complete all additional questions in the submission questionnaire.

Once your full submission is complete, your paper will undergo a series of checks in preparation for re-review. After your manuscript has passed the checks it will be sent out for review. To provide the metadata for your submission, please Login to Editorial Manager (https://www.editorialmanager.com/pbiology) within two working days, i.e. by Oct 20 2022 11:59PM.

Kind regards,

Roli

Roland Roberts, PhD

Senior Editor

PLOS Biology

rroberts@plos.org

---

## [Decision Letter · Decision Letter 1]

22 Nov 2022

Dear Dr Gershenzon,

Thank you for your patience while we considered your revised manuscript "Bark beetles locate fungal symbionts by detecting volatile fungal metabolites of host tree resin monoterpenes" for publication as a Research Article at PLOS Biology. This revised version of your manuscript has been evaluated by the PLOS Biology editors, the Academic Editor, and the three original reviewers.

Based on the reviews, we are likely to accept this manuscript for publication, provided you satisfactorily address the remaining points raised by the reviewers. Please also make sure to address the following data and other policy-related requests.

IMPORTANT:

a) Please make the title more explicit; we also think that it would be helpful to emphasise the importance of the study species - overall, we suggest something like "Conifer-decimating bark beetles locate fungal symbionts by detecting volatile odorants generated by fungal metabolism of tree resin monoterpenes"

b) Please attend to the remaining requests from reviewers #2 and #3. Most of these relate to toning down the strength of the claims, and although reviewer #2's comments are lengthy, s/he is overall very positive, and I believe that the requests are mostly textual and/or presentational in nature.

c) Please address my Data Policy requests below; specifically, we need you to supply the numerical values underlying Figs 1BCD, 2ABCDE, 3A, 4B, 5BCD, 6ABCDEF, 7AB, 8BCD, S1, S2, S3, S4, S5, S6, S7, S8, S9, S10ABC, S11, S12ABCDEFGHIJ, S13BCDEF, S14BCDEF, either as a supplementary data file or as a permanent DOI’d deposition.

d) Please cite the location of the data clearly in all relevant main and supplementary Figure legends, e.g. “The data underlying this Figure can be found in S1 Data” or “The data underlying this Figure can be found in https://doi.org/XXXX”

We expect to receive your revised manuscript within two weeks. 

*Published Peer Review History*

*Press*

Sincerely,

Roli Roberts

Roland Roberts, PhD

Senior Editor,

rroberts@plos.org,

PLOS Biology

DATA POLICY:

Regardless of the method selected, please ensure that you provide the individual numerical values that underlie the summary data displayed in the following figure panels as they are essential for readers to assess your analysis and to reproduce it: Figs 1BCD, 2ABCDE, 3A, 4B, 5BCD, 6ABCDEF, 7AB, 8BCD, S1, S2, S3, S4, S5, S6, S7, S8, S9, S10ABC, S11, S12ABCDEFGHIJ, S13BCDEF, S14BCDEF. NOTE: the numerical data provided should include all replicates AND the way in which the plotted mean and errors were derived (it should not present only the mean/average values).

SPECIES INDICATED IN THE ABSTRACT? 

- Please note that per journal policy, the model system/species studied should be clearly stated in the abstract of your manuscript. 

We require the original, uncropped and minimally adjusted images supporting all blot and gel results reported in an article's figures or Supporting Information files. We will require these files before a manuscript can be accepted so please prepare and upload them now. Please carefully read our guidelines for how to prepare and upload this data: https://journals.plos.org/plosbiology/s/figures#loc-blot-and-gel-reporting-requirements

DATA NOT SHOWN?

REVIEWERS' COMMENTS:

Reviewer #1:

The authors have addressed all of my concerns and appear to have addressed the comments, suggestions, concerns, and suggested experiments of the other reviewers. 

I enjoyed reading/reviewing the ms. 

Reviewer #2:

The authors have done an amazing job of addressing my concerns with the paper. I strongly recommend publication if the authors can address my further comments and recommendations below, and these can be addressed easily. 

One of my original concerns remains - the authors appear to want to force their data into supporting an overly-simplified narrative regarding the chemical ecology of the beetle/symbiont relationship. Several of the experiments do not support their conclusions but are presented as if they do (discussed below). The paper focuses on attraction when clearly oxygenated monoterpenes can have both attractive and repellent effects (this is mentioned but appears to be ignored in the title and elsewhere). There is no conclusive support for the assertion that Ips typographus utilize fungus-produced oxygenated monoterpenes to locate host tissue colonized by beneficial fungi. I do think something very interesting is going on in this system, but their data are not pointing to a single possible narrative. If the authors amend the paper to interpret and present their excellent experiments in a way that considers all possible explanations and ramifications, I think this work will be an outstanding contribution. 

Here are my comments on the authors' responses to my criticisms:

1) My criticism: Deleterious fungal associates are common, particularly bluestain. Authors' response: some bluestain fungi are beneficial. My response: The authors are correct about the existence of beneficial bluestain fungi; this was my error. However, my point stands that there is no general rule that fungal associates that are carried by the beetle can be assumed to be beneficial. The bluestain O. minus is carried by and highly deleterious to brood development (larvae die) of both bark beetles Dendroctonus frontalis and D. brevicomis and the bluestain Ophiostoma ips is deleterious to Ips grandicollis and Ips calligraphus host colonization and brood development. Ophiostoma ips appears to be beneficial to Ips pini, however, which suggests that generalizations within a fungal genus cannot be made. Too few systems have been studied sufficiently to make a generalization about whether fungal associates are beneficial or deleterious. The authors acknowledge as much and that the ecological relationships between Ips typographus and their fungal associates have not been studied sufficiently to draw conclusions. 

2) My criticism: Deleterious fungi were not studied to see if they also produce oxygenated monoterpenes. Authors' response: A new study looked at a deleterious fungal species (Trichoderma) and found it to produce oxygenated monoterpenes but a different blend than produced by G. penicilliata. In a behavioral bioassay, Trichoderma cultures were unattractive to beetles whereas cultures of a symbiont were. My response: I am extremely pleased that they did this experiment. As a graduate student, I compared volatiles arising from symbionts of D. frontalis and Trichoderma harzianum (in my dissertation but unpublished) and also found different blends of oxygenated monoterpenes. There is no question that Trichoderma is not a symbiont and my personal observations indicate that it is deleterious (the mycelium appears capable of infecting beetle brood pupae), although there is nothing I know of published on the topic. Their choice of a "control" fungal species was perfect, I think. However, their argument about the role of oxygenated monoterpenes being the compounds mediating attraction of bark beetles to fungal cultures hinges on the attraction being to specific blends of oxygenated monoterpenes. This is a reasonable hypothesis, but they provide no evidence for it specifically. Additionally, they provide no evidence that oxygenated monoterpenes are the semiochemicals mediating attraction of the beetles to fungal cultures. The fungi produce other compounds, some of which are listed in their very complete tables in the supplementary materials. 

3) My criticism: Oxygenated monoterpenes have many different sources in the environment including the beetles, and therefore, as a class, they cannot be termed "fungal metabolites". Authors' response (1): Ips typographus do not produce the major oxygenated monoterpenes detected from fungal cultures. My response: This is largely accurate, and in the discussion the authors cover this topic well. If the arguments stay limited to Ips typographus, there isn't a problem. If the discussion is expanded to other bark beetle species, there is substantial overlap between oxygenated monoterpenes of beetles and fungi. Authors' response (2): Other sources exist but the blends produced by the different sources differs, and the beetles are responding to different blends. My response: The authors have no evidence for the latter assertion. It would need support from tests showing that (1) synthetic blends of oxygenated monoterpenes produced by fungi are attractive to Ips typographus, (2) the beetles distinguished among blends of oxygenated monoterpenes, (3) the beetles distinguished blends associated specifically with their fungal symbionts. Their existing behavioral tests don't address these questions. 

4) My criticism: The authors' "fungal metabolites" may not be produced by the fungi alone but possibly also reaction by the infested plant tissue to fungal infection. Authors' response: This is as real possibility, however, testing it is extremely challenging, and our own efforts were inconclusive. My response: I am a little confused about why they think this remains an open question, because their new experiment in which sterile and re-inoculated beetles are introduced into logs addresses this topic well. This particular experiment is very informative and helpful to a range of important questions about bark beetle-host interactions, I and applaud the authors for doing this. I am sorry there was not more replication. 

5) My criticism: The work is not original in stating that fungal development is responsible for oxygenated monoterpenes has been published. Authors' response: "Thus, we conclude that ours is indeed the first study to show that symbiotic ophiostomatoid fungi are the main source of I. typographus-attractive oxygenated monoterpenes in Picea abies bark." My response: In retrospect, I was excessively harsh in my criticism regarding originality. My criticism was based on the fact that several studies have shown that levels of oxygenated monoterpenes - including the major ones identified in the authors' study - increase in the days and weeks following bark beetle attack, and authors have attributed this effect to fungal growth. However, the latter was never demonstrated experimentally. Demonstration that fungi (yeasts) can produce oxygenated monoterpenes had previously occurred only in Leufvén et al., 1987, as indicated by the authors. A fair statement is that the hypothesis is not original, but research into the hypothesis has never before been so thorough (particularly in the revised version of the paper) and involved mycelium-producing fungi. I have changed my mind, and I agree that the work is sufficiently original to merit publication.

6) My criticism: The authors fail to account for the fact that one of the experimental contrasts they use as evidence is not sufficiently controlled, as different media were used for experimental (SBA) and "blank" treatments (PDA), and the result was thus inappropriately used to argue that attraction was due to fungal inoculation in the experimental cultures. Authors' response: Additional tests indicate that the SBA alone is attractive but it is "more reasonable" to conclude that the response was to attractive compounds. My response: It is clear from Fig 1D that the spruce bark itself is attractive (important information), but none of the data interpretation rests on experiments where effects of the medium and fungal treatment are conflated. This is good now. 

7) My criticism: Experiments do not demonstrate that fungal production of attractants/feeding stimulants increases in the presence of hydrocarbon monoterpenes from the host, a conclusion that would require demonstration of a statistical interaction between presence of monoterpene and fungus. Authors' response: There was no statistical evidence of an interaction, but "we could clearly show that the presence of the fungus in a beta-pinene enriched diet resulted in significantly longer tunnel length (Fig. 6B)". My response: This is still a significant problem. Figure 6A (not 6B, authors' typo) and 6C only show that the result in figure 1A (PDA cultures are more attractive with growth of G. penicillita) is also observed when the plates contain SPA with either beta-pinene or bornyl acetate. The attraction can be attributed to the fungus alone, hence there is no evidence that presence of beta-pinene or bornyl acetate has a role in changing the attractiveness of fungal plates. Figures 6E and 6F supports the conclusion that the presence of beta-pinene and bornyl acetate have no effect (the combination of fungus and monoterpene is not more attractive than fungus alone). Hence figures B and D are attempting to explain a phenomenon that was never demonstrated in the first place. This does not mean that the results of 6B and D aren't interesting - however. I think that presentation of results that do not support the authors' thesis is important, but figures 6A, C, E and F provide no support to the statement that "Ips typographus is attracted to oxygenated monoterpenes produced by symbiotic fungi" (the title for Figure 6). An additional problem with the title is that repellency was also observed. The figure title and the included data in the figure must be changed. 

8) My criticism: It is possible that oxygenated monoterpenes could be produced by the host tissue also (second mention of this). Authors' response: This is a difficult topic to explore. Also: "We found species-specific VOC profiles for the different fungi when they colonized spruce bark. These seem more likely to occur if the compounds were made by the fungus. If these compounds were produced by the tree as a defense response, one might expect a very similar response to each of these ophiostomatoid fungi." My response: I am not sure why production of species-specific oxygenated monoterpene blends by the fungi in vitro is relevant to this point. I don't think the existence of variation by itself implies that the variation must have some ecological significance, and, in particular, mediation of host selection by bark beetles. It opens the possibility, but is not an argument in support of it.

9) My criticism: The authors claim that oxygenated terpene levels reached a maximum at 18 days is not supported by the data since a longer interval was not tested. Also, it has been shown that concentrations of oxygenated monoterpenes continue to increase in host bark well after the bark is unsuitable for further colonization (no further attacks would be occurring). Authors' response: Text is changed to indicate that the study was the result of a laboratory experiment and that the fungi might also mediate rejection of exhausted host material. My response: Sounds good. I agree that bluestain fungi are not always bad for bark beetles, but too few systems have been studied to make generalizations. 

10) My criticism: Mass attack by aggressive species of bark beetles happens before the associated fungi are established. Authors Response: True, but only during epidemics. When beetle populations are low, colonization occurs slowly, and thus trees with some fungal colonization are still be attacked. My response: This is true and is an excellent point. However, the implication is that fungal activity can have little if any influence on host selection during epidemics, when bark beetles are inflicting their greatest harm. The authors do address this in the discussion. 

11) My criticism: The olfaction studies are well-done and interesting but provide, at best, weak support for the thesis. They suggest relatively limited olfactory specificity to oxygenated monoterpenes and relatively low olfactory sensitivity to those produced in greatest quantity by fungal cultures. Authors Response: Nine classes of ORNs are sensitive to oxygenated monoterpenes. My Response: As the authors indicate, five are specific to beetle pheromones. Of three ORN classes described as fungi-specific, the verbenone group includes bark beetle pheromones (including cis-verbenol) and compounds with other origins (both trans-verbenol and verbenone can arise from autoxidation of alpha-pinene). The isopinocamphone and trans-4-thujanol groups are harder to explain. Authors Response: Limited numbers of ORNs can nonetheless permit discrimination of compounds. My Response: I agree, it is hard to translate numbers or ORN classes into the capacity to distinguish compounds or blends. 

12) My Criticism: The presence of olfactory receptors to oxygenated monoterpenes is no argument in itself that these compounds mediate attraction of beetles to host trees colonized by fungal symbionts. These compounds may also function in signaling unsuitability, and there is compelling evidence for this. Authors' Response: "Since we showed the profiles of oxygenated monoterpenes are distinct among fungal symbionts, saprophytes, and bark beetles, we suggest that bark beetles possess different classes of OSNs to detect compounds from multiple ecological sources including fungi and can distinguish symbionts versus saprophytes based on their volatile profiles. We have now incorporated these points in the discussion in lines 454 and following, and 578 and following." My Response: Sounds good.

13) My Criticism: Results of the behavioral bioassays of response of beetles to individual monoterpenes is uncompelling due to the large number of tests with only two showing significance and at a single, intermediate dose. Authors' Response: However, the mixture of these metabolites produced from spruce resin precursors is strongly attractive when presented in the composition and at the dose produced by each fungus. Reviewer Response: Where is this experiment in the paper? Authors' Response: However, we could not perform a Bonferroni test to make such corrections here as we only performed a single pairwise non-parametric Wilcoxon's test (choice of beetles between control and treatment) for each experiment. Reviewer Response: The Bonferroni correction can be applied to any number of tests in an experiment, although typically its use is limited to multiple comparisons of treatments in a single experiment. The principal behind the test is still valid in other situations - the more tests that are run, the more likely that one will see a type 1 error, and this likelihood can be calculated by dividing the alpha level by the number of tests. Although statistical practice does not mandate application of a Bonferroni correction to the authors' data, the principal nonetheless influences interpretation of the data and demands skepticism be given to the broad statement that "Ips typographus is attracted to oxygenated monoterpenes produced by symbiotic fungi". There were 21 separate tests looking at beetle response to individual oxygenated monoterpenes (figure 6B and D), hence the corrected alpha-level is 0.0023 which indicates that there was a high probability that the observed positive responses were the result of chance. The results are interesting but are not a sufficient basis for asserting a new paradigm for bark beetle-fungal interactions. I think it can stay as-is. 

14) My criticism: The sum total of problems merits rejection. Authors' Response: We will revise the paper to address your concerns. Reviewer response: I greatly appreciate the extensive work taken to address my concerns. 

L23 Is 3-6 mm the true size range?

L23 Insert "possible" before role. The sentence implies the existence of such is already known or commonly hypothesized.

L29-31 Change "We previously demonstrated…" to "Evidence indicates…" 

L29 The word "recognize" is too vague and suggests much more than the evidence support. Be specific. (e.g. "can distinguish cultures of"). 

L29 There are no studies demonstrating that any of the fungi in this study were beneficial or not. Statements to this effect must be removed. They are symbionts of unknown ecological relationship. Remove "beneficial". 

L35 One cannot say "converting" without experiments that would involve labelled precursor. The oxygenated monoterpenes appear when the fungus is in the presence of resin monoterpenes. That "conversion" is occurring is likely but not proven. It is proven that hydrocarbon monoterpenes + fungal growth = appearance of oxygenated monoterpenes, but this is not a demonstration of metabolism, although it seems the most obvious hypothesis for the result. 

L34-38 The language here suggests that the pathway has been identified (generally one does not draw conclusions about metabolism without a pathway being demonstrated). The wording should be changed to reflect that the data showed input of one compound resulted in the appearance of the other and no more than this. De novo production has not been ruled out [Leufven et al (1988) found oxygenated monoterpenes appearing in their yeast cultures even though autoclaving of their media had appeared to have volatilized all hydrocarbon monoterpenes] although oxidation is the simplest explanation. Another alternative explanation is that the fungi create an environment where spontaneous oxidation is more likely (e.g., by increasing exposure to oxygen or by providing surface area for oxidation to occur), although the appearance of species-specific blends is not consistent with these hypotheses. Please state that the data "suggest", "imply" etc. a mechanism unless this mechanism has been demonstrated explicitly. 

L38-39 There is NO evidence in the paper to support the statement that Trichoderma produced oxygenated monoterpenes in "non-attractive ratios"! Trichoderma is likely non-beneficial but there is no evidence to this effect. It is safe to say it is not a symbiont, or a non-symbiotic fungus that occupies the same habitat as the symbionts. I recommend the authors seek papers discussing Trichoderma's association with bark beetle killed trees.

L39 Delete "extensive"

L41 Delete "such as"; camphor and thujanol were the only ones. 

L42 Change "olfactory" to "walking olfactometer". 

L47 Remove "essential" unless you can cite published papers that demonstrate that this is the case. The ecological relationship is unknown, but could include mutualism, commensalism, and antagonism. Since it is shown that oxygenated monoterpenes can be repellant to the bark beetles, I think it makes for a more interesting narrative to discuss the possibility that the different fungi producing the oxygenated monoterpenes could be deleterious or beneficial. There is some mention of this in the discussion, but I would like this to be fully embraced by the paper. 

L79 Delete comma and replace "which" with "that"

L80 Replace "raises broods" with "reproduces"

L85-86 Change sentence. The ecological relationships between these fungal species and I. typographus have not been extensively studied. 

L88. Add sentence "However, some ophiostamatoid associates of bark beetles have been shown to be commensals or antagonists to beetle brood development."

L98 Change "Interestingly," to "Additionally, "

L103-105 Clarify this. There are quite a few studies that have looked at the composition of resin within lesions on conifers following inoculation with bark beetle fungal associates. Generally these studies didn't look for oxygenated monoterpenes, but the presence of the fungi did result in a change in volatile extractables from the phloem. 

L106-107 Insert "excised" before "bark". 

L108 Remove "dramatically" 

L111 Change "indicate" to "suggest" or "imply". This statement requires a lot of untested assumptions, but it is a scenario that would explain the results. 

L121 Delete "much"

L123. Explain "primary" and "secondary" in the introduction. What does this mean and what evidence is used to make these classifications? 

L124 Delete "highly". 

L125 Identify Trichoderma as a possibly deleterious, non-symbiont and provide citations to support this. 

Table S2 GC-FID is not a sufficient means for identification when chromatograms are as complex as the ones in this study. My experience is that a large amount of erroneous data is introduced into the literature when this is done, and the practice is not acceptable in chemical ecology journals. The authors need to defend their approach here. 

Figure S2. Indicate the number of compounds used in the analysis.

L135 The figure/table references are unclear. 

L138 Delete parenthesis after "monoterpenes"

L141 Insert "for dates sampled" after "reaching a maximum"

L142-143 Figure 2B suggests just the opposite, with the mean proportion of oxygenated monoterpenes increasing several-fold. Table S3 supports the statement. You should give the statistics for the uninoculated treatment and mention something about the variance or in some way reconcile the text and the figure. The P-value for G. penicillata seems strangely high (on the edge of significance) given the enormous difference in means (~10 fold). Consider adding statistics for summed compounds of each class in tables S3-S7. 

L146-148. Again, the data represented in figure 2B seem inconsistent with these stats. In 2B, the change in proportion of hydrocarbon monoterpenes was only 7% whereas the P-value was very low. Make a comment to reconcile this

Figure 2. Be consistent about placing the reference lettering either before or after the associated text. Remove the sentence "Principal…treatment" as this information is apparent in the figure. 

L146 If the Tukey test results aren't reported here, delete "Tukey's test"

L158 Move "(Fig…Table)" before "including"

L163 Normally "Fig" is written before the associated number/letter. Be consistent about this. 

L166 Bornyl acetate is technically an "oxygenated monoterpene". The authors restrict the use of "oxygenated monoterpene" to alcohols, ketones, and aldehydes. Some comment on this limitation would be helpful. 

L173 Insert a comma after "similar"

L175 "included." Change "from" to "in the presence of"

Table S2 GC-FID is not a sufficient means for identification when chromatograms are as complex as the ones in this study. My experience is that a large amount of erroneous data is introduced into the literature when this is done, and the practice is not acceptable in chemical ecology journals. The fact that large amounts of spiroketals (fairly rare compounds) were detected in large quantities in this FID study, but none or small amounts in the GC-MS studies, increases my skepticism. 

L186 Remove the word "dramatically". 

L192-200. Examination of an insect with electron microscopy as not compelling evidence of sterility. There are many concealed locations on the exoskeleton of an insect where propagules could hide, and SEM obviously would not detect propagules carried internally. The plating data (Table S9) are the only meaningful evidence of removal of mycelial fungi. Furthermore, figure 4A does not support the assertion that fungi were successfully reintroduced to the cuticle. Figure Aii clearly shows classic "pillow shaped" ophiostamatoid sexual spores, whereas 4A shows nothing that looks like a fungal propagule. Hence, 4A provides no support (but does not contradict) the contention that fungi were successfully re-introduced, and should probably be removed. 

L206 Missing a parenthesis

203-206 In figure 4, reinoculated beetles were not significantly different from fungus free beetles. For terpinene-4-ol, there was no difference between no beetles and the reinoculated beetles. Reconcile this.

189-222 Despite the difficulties with re-inoculated beetles, this is still a very compelling experiment I think, and it is an outstanding contribution to the literature. 

L271 The statement "within the range of natural concentrations in P. abies bark" requires support from experiments or citations.

L277-278 The most obvious explanation for the third test in 6A is that beta-pinene was causing the repellancy. Why was it presumed that an oxygenated monoterpene was doing this?

L285-287 This conclusion is NOT supported by the results. There is no evidence that the combination of beta-pinene and fungus is more attractive than fungus alone, and there is no evidence that repellency is caused by a metabolite of beta pinene and not beta-pinene itself. There is evidence that specific oxygenated monoterpenes demonstrated elsewhere to be produced by the combination of fungi and host tissue can be either attractive or repellant to beetles. This statement as well as the title of the section must be changed to reflect this. 

L303-305 This statement belongs in the methods.

Figure 7. The figure is a bit confusing because the compound combinations are indicated on the left hand side of the axis. I am not sure how to rearrange this but, one gets the impression that the contrast is between the synthetic pheromone and the combinations. Replacing "synthetic pheromone" with "pheromone components" would be helpful.

L310 Delete "significant"

L315 Consider deleting "success"

L320 The reference to units isn't helpful. Just state the percentages.

L315-338 These are good data, but tests of interactions between monoterpene presence and fungus would be informative regarding the hypothesis regarding the possible role of oxygenated monoterpenes. 

L337 figure 8B is the only evidence in the paper indicating that the combination of a terpene and fungus is more attractive than either singly, a finding supportive of the oxygenated monoterpene hypothesis. 

L342 Entomocorticium are badiomycetes.

L350 Data is presented that these compounds increased, but not they became dominant. 

L352. There is no evidence presented that oxygenated monoterpenes stimulate mining. 

L388-391. This cannot be concluded from the data in this paper. The data presented do not demonstrate that oxygenated monoterpenes produced by fungal growth are attracting beetles to fungal cultures. There are no data presented on blends. Only two compounds (one of the two - trans-4-thujanol - is produced by G. pinicillata in only minute amounts) were shown to be attractive singly, and no blends were tested. Acceptance of the blends hypothesis would only be justified by tests with synthetic blends matching those produced by the fungi in which identical results were observed as with the fungi themselves. The fact that the oxygenated monoterpenes have many sources including non-symbionts is a problem for the hypothesis that oxygenated monoterpenes are used by beetles to distinguish beneficial fungi, no question, but to accept the blends hypothesis as the only possible solution is unjustified. The role of oxygenated monoterpenes in allowing beetles to distinguish fungi has not been demonstrated definitively in this study, hence there is no urgency in resolving an inconsistency conflicting with this unproven hypothesis. The blends hypothesis must be indicated as one possible explanation. Another possible explanation (and the simpler one) is that oxygenated monoterpenes aren't what is allowing insects to distinguish fungi, and the variation in oxygenated monoterpene blends is irrelevant. Relevant example: Healthy conifers typically have very specific blends of resin monoterpenes both among species and geographically, and many of these monoterpenes are attractants for bark beetles. However, I am not aware of evidence that the beetles discriminate blends. Is it acceptable to assume that, merely because there is variation in terpene blends associated with conifers, that bark beetles must be highly sensitive to blends, and there is no need to test this hypothesis? 

L391-395 Since the blends hypothesis is not demonstrated or supported by tests of its validity, its existence as a known phenomenon cannot be used in the discussion. 

L407 "Metabolism" is perhaps too specific. It could be a signal that defenses have been overcome or reduced, however the fungi accomplish this.

L416 Insert "could" before "provide"

L434 Delete "natural attacks upon"

L436 Change "within few days and" to "sufficiently rapidly to"

L437-439 I don't see where it was demonstrated that specifically oxygenated monoterpenes increased tunnelling. It is one possibility, and there are others. Modify this statement accordingly. 

L440 Was there a reason to suspect there would be sex-specific differences? Why was sex specificity only examined in the tunnelling and pheromone experiments?

L451 Delete comma

L452 Verbenone can also be produced by autoxidation directly from alpha-pinene. 

L454-455 I suggest deleting this sentence, since obviously structural differences produce different behaviors, and the same compound does not have a different effect at different stages of the beetle life cycle. 

L457 Insert "natural" before "enemies"

L462-465 There is no evidence that this is mediated by oxygenated monoterpenes.

L456-471 Other relevant citations: Salom, S.M., Ascoli‐Christensen, A., Birgersson, G., Payne, T.L. and Berisford, C.W., 1992. Electroantennogram responses of the southern pine beetle parasitoid Coeloides pissodis (Ashmead)(Hym., Braconidae) to potential semiochemicals. Journal of Applied Entomology, 114(1‐5), pp.472-479. Salom, S.M., Birgersson, G., Payne, T.L. and Berisford, C.W., 1991. Electroantennogram responses of the southern pine beetle parasitoid Dinotiscus dendroctoni (Ashmead)(Hymenoptera: Pteromalidae) to potential semiochemicals. Journal of chemical ecology, 17(12), pp.2527-2538.

L504-505 How did you identify it as a metabolite of alpha- and beta-pinene?

L506 Replace "another" with "an"

L506-507 Provide citations

L522-524 I suppose exposing wounds to dead tissue of different fungal species (this can be sufficient to elicit a defensive response) might be a way to get at this. There is evidence that feeding by certain insect herbivores elicit species-specific secondary volatiles from their hosts, and these blends have been hypothesized to attract specialist naturally enemies. So it is not impossible that the tree might respond with specific blends. You don't need to mention this…

L559 What kind of insect is Xenopus?

L566-569 Nice

L586. There was no evidence presented that metabolism was how the compounds arise, only that the presence of fungi increased their concentrations. 

L589 There was no evidence presented that oxygenated monoterpenes had any effect on beetle attraction to fungi or tunneling. Clearly the fungi have an effect on attraction, but none of the experiments showed that fungal oxidation of monoterpenes played any role.

L625 Were there steps to keep the phloem cold or otherwise from oxidation etc. prior to grinding?

L636-649 It doesn't seem that steps were taken to preserve the sterility of the collection bottles. It seems very unlikely that incidental microbes didn't start to grow on the exposed tissue of the controls. I think it would be better to say "uninoculated" rather than "uninfected" since the microbial state of the control plugs was unknown. 

L660-662 How were quantitations performed with the GC-MS? Were standard curves produced? L674 I am not familiar with this method of volatile collection, but it appears to involve an adsorbent placed in a static headspace and the desorbed thermally. Adsorbents tend to be selective and thus the proportions of compounds collected may not be representative of the proportion in the headspace. Also, accurate quantitation is time consuming (requires calibration of the sampling device). How was all of this addressed? 

L693-694 It seems that the monoterpenes were added while the PDA was still hot enough to pour. Was anything done to compensate for evaporation of the monoterpenes? Was the 0.5 mg/ml likely to be accurate?

L764-778 The construction of the olfactometer is not sufficiently clear from the text. Figure 1A provides some information, but it needs to be in the text. A detailed and labelled figure would be helpful. Photographs in the supplementary materials would be helpful also. It is not explained (1) How the beetles were able to get into the cups (there are holes evident in Fig 1A - how many, how large, and how high off the floor? What material were the cups made of? Were these purchased or hand-constructed? (2) What did the larger arena (containing the cups) consist of (material, diameter, height)? (3) Small holes are apparent in the walls of the larger arena in Fig 1A - what are the details of these and what was their purpose? (4) L268 states that two beetles were placed inside each arena - how many bioassays were done at one time? (5) Were the beetles actually "trapped" (couldn't escape) or were they merely inside the cups when they were examined? (6) "For up to six hours" - the cups were checked multiple times? What determined when the assay was over? How was each successive observation of the same arena included in the final data set? (7) At what temperature and humidity were the assays conducted? (8) When the beetles were placed into the arena, where were they placed? How were they released (with a forceps, etc.)? (9) Why two beetles per assay? (10) Were beetles re-used? (11) Were the arenas/cups re-used, and, if so, how were they cleaned between uses? (12) If known, what was the air flow rate in the laminar flow hood (high air flow could interrupt with semiochemical responses)? (13) Was the arena covered in some way? (14) Was treatment assignment to cups randomized? 

This information is all necessary to ensure that the bioassays described are repeatable

L825 It is not clear what "combined" means. 

L833-834 The behavioral bioassays were binary data—how was a Wilcoxon's test applied to these?

L834 "signed"

Reviewer #3:

This revision has some substantial improvements over the original version. In addition to some revised wording the authors conducted additional experiments that demonstrate specificity by the host insect toward fungal volatiles. That is very helpful. The revised manuscript does, however, require additional improvement in two areas. First, as I pointed out in the original and Reviewer 2 also indicated, some of these experiments do not account for experiment-wise error. I appreciate and agree with the authors' response that their experimental design does not allow for such corrections, but that does not exempt them from the constraints on inference that such non-accounting impose. In particular, I do not see how general trends can be further delineated into responses purportedly going up, down, or both with specific doses without either fitting an overall dose-response curve or explicitly accounting for experiment-wise error. So those statements should be adjusted to what inferences can actually be made from the experimental design. Second, the authors now do a better job than before of relating these experiments to the natural condition of beetles arriving at fallen trees, but at times the language appears to encompass the rapid mass attacks by which these beetles overwhelm the defenses of healthy live trees. The experimental design and time frame are not applicable to the latter.

There are also a few remaining additional issues that can be easily corrected with minor rewording:

1. Line 88: Reference 20 is about bacteria, not fungi. So that needs to be stated separately rather than within a description of ectosymbiotic ophiostomatoid fungi.

2. Line 402: It would be good to insert 'visual and chemical' in front of 'signals' to inform the reader that beetle orientation involves multiple behavioral modalities.

3. Line 442: 'beneficial to both sexes'. This should be reworded to make more clear whether it refers to feeding by male and female adults or by larvae that develop into males or females.

4. Line 495: 'Whether bark beetles, their associated fungi, or other microbes can detoxify diterpenes is not yet known' This was shown for bacteria in Boone et al. 2013.

5. Line 515. A useful reference to cite here is Hunt et al. 1989. They did observe some auto-oxidation of alpha-pinene, but in agreement with your findings it was very low and very delayed: only 0.8% and 1.9% of the trans-verbenol produced by D. ponderosae and D. rufipennis respectively, and over a time span too lengthy to significantly influence mass attack.

6. Line 580" What are 'higher' centers of the brain? Reword.

---

## [Editor Report · Decision Letter 2]

12 Jan 2023

Dear Jonathan,

Thank you for the submission of your revised Research Article "Conifer-killing bark beetles locate fungal symbionts by detecting volatile fungal metabolites of host tree resin monoterpenes" for publication in PLOS Biology. On behalf of my colleagues and the Academic Editor, Anurag Agrawal, I'm pleased to say that we can in principle accept your manuscript for publication, provided you address any remaining formatting and reporting issues. These will be detailed in an email you should receive within 2-3 business days from our colleagues in the journal operations team; no action is required from you until then. Please note that we will not be able to formally accept your manuscript and schedule it for publication until you have completed any requested changes.

Sincerely, 

Roli

Senior Editor

PLOS Biology

rroberts@plos.org